# DNA origami-based single-molecule force spectroscopy elucidates RNA Polymerase III pre-initiation complex stability

Kevin Kramm[1], Tim Schröder [2], Jerome Gouge[3], Andrés Manuel Vera [2], Kapil Gupta [4], Florian B. Heiss [5], Tim Liedl [6], Christoph Engel [5], Imre Berger [4], Alessandro Vannini [3,7], Philip Tinnefeld[2] & Dina Grohmann [1,5✉]

The TATA-binding protein (TBP) and a transcription factor (TF) IIB-like factor are important constituents of all eukaryotic initiation complexes. The reason for the emergence and strict requirement of the additional initiation factor Bdp1 in the RNA polymerase (RNAP) III system, however, remained elusive. A poorly studied aspect in this context is the effect of DNA strain arising from DNA compaction and transcriptional activity on initiation complex formation. We made use of a DNA origami-based force clamp to follow the assembly of human initiation complexes in the RNAP II and RNAP III systems at the single-molecule level under piconewton forces. We demonstrate that TBP-DNA complexes are force-sensitive and TFIIB is sufficient to stabilise TBP on a strained promoter. In contrast, Bdp1 is the pivotal component that ensures stable anchoring of initiation factors, and thus the polymerase itself, in the RNAP III system. Thereby, we offer an explanation for the crucial role of Bdp1 for the high transcriptional output of RNAP III.

[1] Single-Molecule Biochemistry Lab, Institute of Microbiology and Archaea Centre, University of Regensburg, 93053 Regensburg, Germany. [2] Department of Chemistry and Center for NanoScience (CeNS), Ludwig-Maximilians-Universität München, 80539 München, Germany. [3] Division of Structural Biology, The Institute of Cancer Research, London SW7 3RP, UK. [4] Bristol Synthetic Biology Centre BrisSynBio, Biomedical Sciences, University of Bristol, 1 Tankard's Close, Clifton BS8 1TD, UK. [5] Regensburg Center of Biochemistry (RCB), University of Regensburg, 93053 Regensburg, Germany. [6] Faculty of Physics and Center for Nanoscience (CeNS), Ludwig-Maximilians-Universität München, 80539 Munich, Germany. [7] Human Technopole Foundation, Centre of Structural Biology, 20157 Milan, Italy. ✉email: dina.grohmann@ur.de

                                                                                    1

All cellular life depends on the regulated expression of its genome. The first step in gene expression is transcription carried out by highly conserved multisubunit RNA polymerases (RNAPs) that use a DNA template to synthesise RNA[1]. Transcription is a cyclic process that can be divided into the initiation, elongation and termination phases. Aided by several general transcription factors (GTFs), eukaryotic RNAPs are recruited to the promoter DNA, thereby positioning the RNAP at the transcription start site (TSS)[2,3]. All eukaryotic RNAPs rely on the GTFs TATA-binding protein (TBP) and a TFIIB-like factor[4–7]. TBP is highly conserved in structure and function and recognises an AT-rich DNA stretch, the so-called TATA-box (eukaryotic consensus sequence TATAWAWR with W = T or A and R = G or A[8]), upstream of the TSS[9–13]. Canonical binding of TBP to the DNA invokes a 90° bend in the DNA[14–16] when two conserved pairs of phenylalanines are inserted into the DNA between bases 1/2 and 7/8 of the TATA-box sequence. TFIIB-like factors associate with the TBP–DNA complex via the C-terminal core domain and concomitantly recognise the B-recognition element (BRE) adjacent to the TATA box[17–23]. Even though additional factors (e.g. TFIIE, TFIIH, TFIIF, TFIIA) are involved in the initiation process in vivo[24], the minimal configuration of TBP and TFIIB is sufficient to recruit the RNAP (in complex with TFIIF) to the promoter in the eukaryotic RNAP II transcription system[25–29]. The auxiliary GTF TFIIA was shown to stabilise the TBP–DNA complex[30,31], thereby stimulating basal transcription levels[32]. Structural studies showed that TFIIA binds the convex side of TBP and forms interactions with the DNA upstream of the BRE/TATA element[33,34]. While the eukaryotic RNAP II system is responsible for the transcription of messenger RNAs (mRNAs) and small nucleolar RNAs (snRNAs), RNAP transcription systems I and III are transcribing ribosomal RNAs (rRNAs) and 5S rRNA, U6 snRNA, transfer RNAs (tRNAs), respectively. The initiation factor setup in the specialised RNAP I and III transcriptions systems, however, diverged from the composition of the RNAP II system and additional initiation factors are required for efficient initiation[5,21,35]. While TBP was found to be part of the RNAP I initiation machinery in vivo[36–38], basal transcriptional activity can be achieved in the absence of TBP[39–41] and its functional role in the RNAP I system remains elusive. RNAP III transcription is directed from three different promoter classes that differ in promoter elements and initiation factor requirement[7,42]. In all cases, transcription initiation relies on the multisubunit factor TFIIIB consisting of TBP, the TFIIB-like factor Brf1 and Bdp1[5,20]. Bdp1 is unique to RNAP III transcription initiation and has no homologue in the RNAP I or II transcription system. Bdp1 is crucially involved in promoter recognition and DNA opening[43,44]. Vertebrates additionally use a TFIIIB variant containing Brf2 instead of Brf1. Both factors are structurally similar, but Brf2 binding to the TBP/DNA complex is regulated by the redox state of the cell[45]. The Brf2-containing TFIIIB complex initiates transcription at a small subset of genes, including the selenocysteine tRNA and U6 snRNA. In contrast to RNAP II-transcribed snRNA genes, the U6 promoter contains a TATA box that is crucial for the specific recruitment of TFIIIB[43,46,47]. TFIIIB is sufficient for the recruitment of yeast RNAP III in vitro[48]. However, at human type 3 promoters an additional protein complex is involved in transcription initiation, the snRNA-activating protein complex (SNAPc, reviewed in ref. [42]).

In addition to biochemical and structural studies, single-molecule Förster resonance energy transfer (smFRET) and ensemble kinetic studies provided insights into the molecular mechanisms and kinetics of transcription initiation[30,49–57]. The interaction of human TBP with the U6 promoter is characterised by a lifetime in the millisecond range[57], while interaction of yeast TBP with an RNAP II or a U6 RNAP III promoter is highly stable

for minutes to hours, and bending occurs in two steps[49,58]. TFIIB and TFIIA were shown to increase the lifetime of the bent state in the RNAP II system[30,49]. Similar to TFIIB, the TFIIB-like factor Brf2 prolongs the lifetime of the TBP–DNA complex[57].

Transcription assays as well as smFRET-based DNA bending assays are performed using short naked double-stranded DNAs (dsDNAs). In vivo, however, transcription initiation factors assemble on the promoter DNA in the context of compact nucleosome structures. As a consequence, the transcriptional landscape in eukaryotes is shaped by chromatin remodelling events[59]. A number of studies analysed the effect of the nucleosome positioning on transcriptional levels and demonstrated that accessibility of the promoter DNA correlates with transcriptional efficiency[60]. Another regulative aspect of the nucleosome organisation is the topological effect on DNA introduced by tightly spaced nucleosomes[61] and the transcription (and replication) machinery. In this context, DNA is subject to mechanical forces. The effect of these forces on transcription initiation, however, has not been analysed as suitable methodological tools were not available so far. Standard force-sensitive methods like magnetic and optical tweezers require long DNA linkers that connect the DNA under investigation to the macromolecular world. In tweezer experiments, a topological change of the investigated DNA can only be transmitted to the beads via this linker, which introduces a considerable noise. Consequently, subtle changes in DNA topology introduced by DNA-binding proteins like TBP are difficult to detect[62].

Here, we utilise a recently developed DNA origami-based force clamp[63] to monitor the influence of DNA strain on the assembly of GTFs from the human RNAP II and RNAP III transcription system on the promoter DNA. Our data establishes Bdp1 as the pivotal component of the RNAP III initiation complex that ensures stable anchoring of the initiation factor TFIIIB, and of the RNAP III at the promoter. This exceptional stability provides a stable anchor point for RNAP III at the promoter, which supports transcription of the short U6, tRNA and 5S rRNAs. Moreover, we demonstrate that the DNA origami force clamp is a powerful tool to study the force dependency of complex protein assemblies and that detailed mechanistic and kinetic information about biological processes can be derived.

## Results

**DNA origami-based force spectroscopy of initiation complexes.** Recently, we introduced a DNA origami-based force clamp that exerts forces in the piconewton (pN) regime on a DNA segment (Fig. 1a)[63]. This nanosized force clamp exploits the entropic spring behaviour of single-stranded DNA (ssDNA) that is placed in the middle of the DNA origami clamp. Forces are tuneable by adjusting the length of the ssDNA that is connected to the rigid body of the DNA origami, thereby providing two fixed anchor points for the ssDNA (Fig. 1b). Due to the reduced conformational freedom of a short DNA segment (equivalent with a reduced entropy of the system), higher strain (e.g. force) acts on the DNA. The resulting forces were calculated using a modified freely jointed chain model[63,64] (for details see Supplementary Information). In this study, we employed DNA origami force clamps with forces ranging from 0 to 6.6 pN. The major advantage of the nanoscopic force clamp is that it acts autonomously and does not require a physical connection to a macroscopic instrument. Moreover, the DNA origami force clamp can be produced and used in a highly parallelised manner. In order to study the force dependency of transcription initiation factor assembly on the promoter DNA, we engineered a prototypical RNAP II (Adenovirus major late promoter, AdMLP) and RNAP III promoter (human U6 snRNA promoter) sequence into the

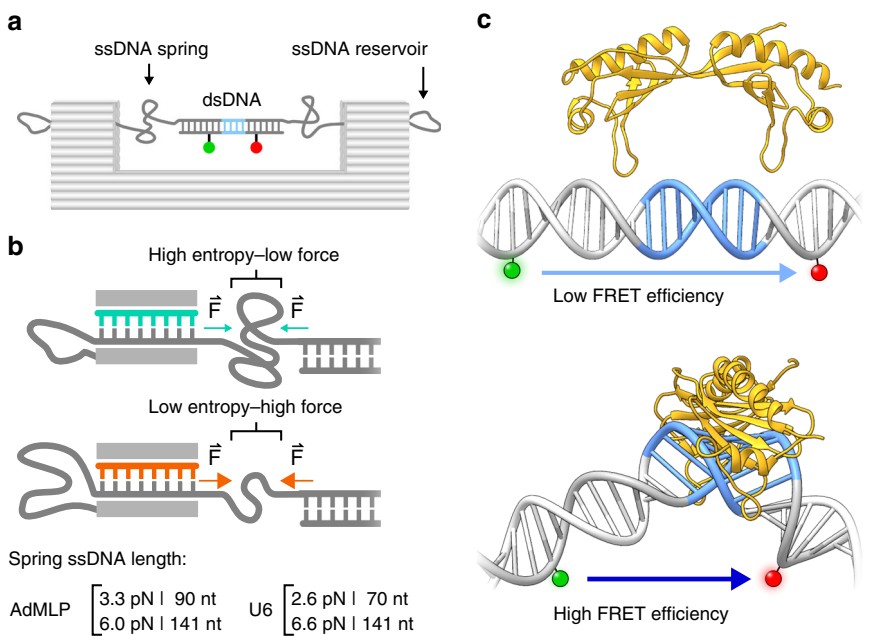

**Fig. 1 DNA origami-based force clamp monitors TBP-induced DNA bending under force. a** Schematic overview of the DNA origami force clamp. The ssDNA spring protrudes from the DNA origami body and spans the 43 nm gap of the rigid DNA origami clamp body (grey). Centred within the ssDNA spring is a double-stranded promoter region including the TATA-box element (blue) flanked by a donor/acceptor (green/red) fluorescent dye pair for FRET sensing. **b** The ssDNA spring length can be adjusted with DNA from the reservoir by using different staple strands (teal/orange) during assembly. Reducing the number of nucleotides spanning the gap leads to a smaller number of adoptable conformations of the ssDNA chain and thus results in a higher entropic force. The length of the ssDNA spring for low (AdMLP: 3.3 pN/U6 promoter: 2.6 pN) and high forces (AdMLP: 6.0 pN/U6 promoter: 6.6 pN) is shown. **c** Single-pair FRET assay as readout for the bending of promoter DNA by the TATA-binding protein (TBP, yellow). A donor (ATTO 532, green) and acceptor fluorophore (ATTO 647 N, red) flank the TATA-box element (blue), resulting in a low efficiency FRET between both dyes. Binding of TBP bends the DNA by ~90°, thereby decreasing the distance between the fluorophores resulting in an increase in FRET efficiency (DNA/TBP structures adapted from PDB: 5FUR).

DNA origami (Supplementary Fig. 1). The AdMLP contains a TATA-box and BRE element sequence, which are targeted by TBP and TFIIB, respectively. The TATA box of the U6 snRNA promoter is flanked by the GR element at position −3/−4 and TD motif at position +3/+4 relative to the TATA box (Supplementary Fig. 1), which are bound by the TFIIB-like factor Brf2[57]. Annealing of a short complementary additional DNA strand that carries a donor (ATTO 532) and acceptor (ATTO 647N) fluorophore allows the detection of TBP-induced DNA binding via smFRET measurements (Fig. 1c and Supplementary Fig. 1). The correct folding of the DNA nanostructure was verified using transmission electron microscopy (Supplementary Fig. 2). The successful hybridisation of the fluorescently labelled DNA strand is demonstrated by fluorescence correlation spectroscopy measurements as the short dsDNA promoter diffuses seven times faster than the respective DNA origami where the labelled DNA is part of the high molecular weight DNA origami structure (Supplementary Fig. 3). We first performed smFRET measurements on freely diffusing DNA origamis and found a single uniform low FRET population for all forces for the AdMLP and U6 promoter force clamps (Figs. 2 and 3). The measured FRET efficiencies are in good agreement with FRET efficiencies obtained from linear dsDNA promoter DNAs (Supplementary Figure 4). This demonstrates that the conformation of the promoter DNA is not significantly changed when it is incorporated into the DNA origami force clamp and forces are applied.

**TBP-induced promoter DNA bending under force.** First, we probed the force dependency of the human RNAP II transcription initiation complex formation. Basal transcription levels in the RNAP II transcription initiation can be achieved using TBP and

TFIIB only. Hence, we added TBP or TBP/TFIIB to the DNA origami force clamp that carries a canonical RNAP II promoter (AdMLP). At the TBP concentration chosen (20 nM), 50% of the molecules showed a high FRET value with a FRET efficiency of 0.64 at 0 pN (Fig. 2, Supplementary Fig. 5 and Supplementary Table 2). Similar results were obtained using linear dsDNA demonstrating that the DNA origami force clamp is suited to probe TBP-induced DNA bending (Supplementary Fig. 4). An increase in force to 3.3 and 6.0 pN resulted in a decrease in the fraction of the high FRET population with only 15% of the molecules in the high FRET state at 6.0 pN (Fig. 2, Supplementary Fig. 5 and Supplementary Table 2). These data show that the bending of a RNAP II promoter by TBP is force dependent. Similarly, the addition of human TBP to the U6 promoter DNA origami led to the appearance of a high FRET population ($E = 0.39$), while the fraction of the U6 promoter bent by TBP is reduced at higher forces (Fig. 3; these data will be discussed in detail below).

**TFIIB and Brf2/Bdp1 establish stable initiation complexes.** As TFIIA was shown to stabilise the TBP-DNA interaction[30], we first examined whether TFIIA exerts a stabilising effect. Indeed, compared to the TBP/AdMLP DNA complex, the addition of TFIIA slightly increased the high FRET fraction at all force tested (Fig. 2, e.g. 15% high FRET without TFIIA at 6.0 pN versus 26% high FRET in the presence of TBP and TFIIA). The addition of TFIIA did not affect the FRET efficiency of the bent state, suggesting that TFIIA does not influence the bending angle of the TBP/DNA complex. In contrast, TFIIB increased the high FRET state fraction more efficiently than TFIIA. At 0 pN 73% of all molecules were found in the high FRET state (Fig. 2). Increasing

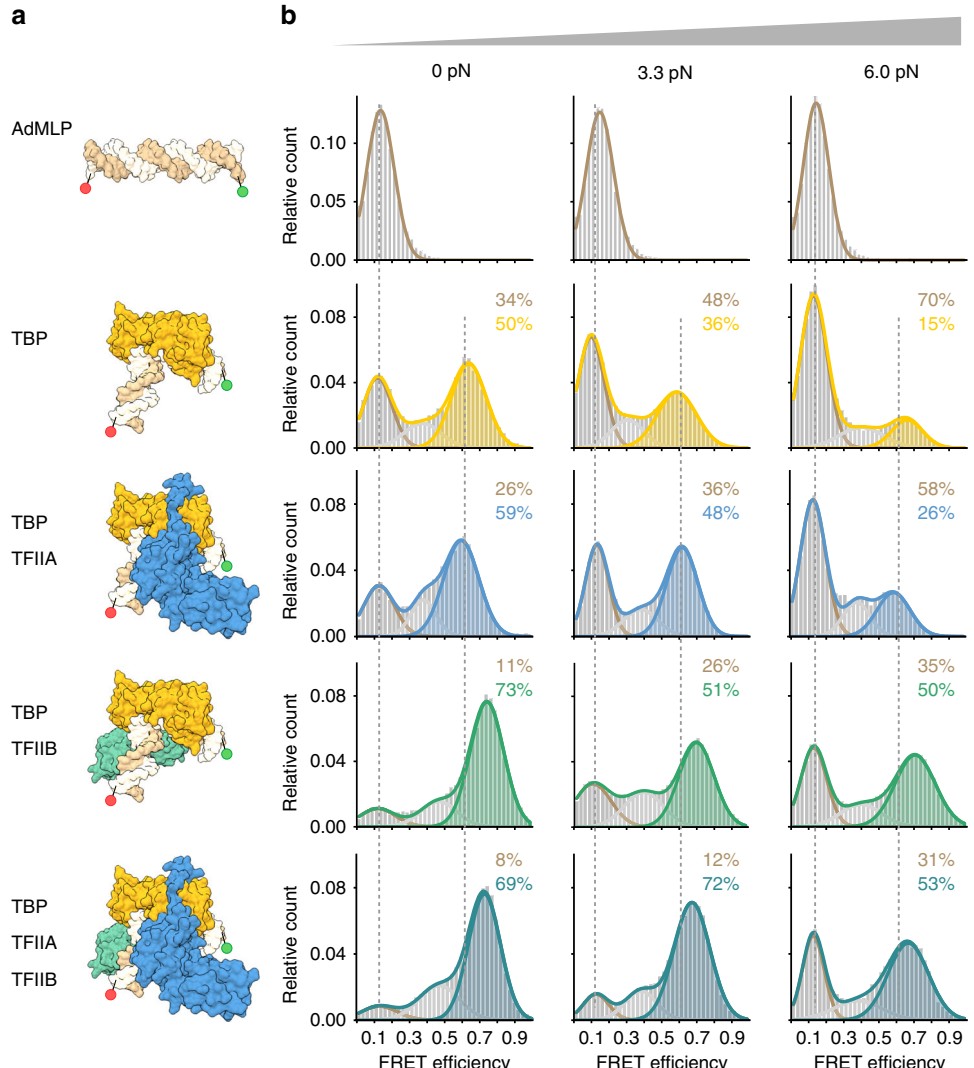

**Fig. 2 Force dependency of promoter binding of RNAP II initiation factors at a canonical RNAP II promoter. a** Structural models (PDB: 5IYB) of the adenovirus major late promoter (AdMLP, brown) in an unbent conformation and the bend state bound by TBP (yellow), TFIIA (blue) and TFIIB (green). **b** Single-molecule FRET measurements on diffusing molecules monitor TBP-induced DNA bending after the addition of TBP (20 nM), TBP and TFIIA (2 μM), TBP and TFIIB (200 nM) or TBP, TFIIA and TFIIB to the AdMLP DNA origami force clamps at increasing forces (0, 3.3 and 6.0 pN). FRET efficiency histograms showing the relative distribution between the unbent DNA state (low FRET state, $E = 0.12$, brown) and TBP-induced bent state (high FRET population, $E = 0.63$ (TBP only and TBP/TFIIA, yellow and blue, respectively), $E = 0.72$ (TBP/TFIIB, green and TBP/TFIIA/TFIIB, teal)). Low and high FRET populations were fitted with a Gaussian distribution. Each measurement was carried out at least three times. See also Supplementary Fig. 5 and Supplementary Table 2.

the force to 3.3 and 6.0 pN resulted in a decreased high FRET population. However, at 6.0 pN a significantly higher fraction of molecules (50%) exhibited a high FRET state as compared to the samples that only contained TBP. Moreover, the high FRET is shifted to a value of $E = 0.72$ indicating that the bending angle is slightly increased in the presence of TFIIB. These results suggest that TFIIB significantly stabilises the TBP-DNA interaction, which is in agreement with previous smFRET studies that showed that TFIIB not only extends the TBP/AdMLP DNA complex lifetimes, but also shifts the equilibrium towards the fully bent state[49]. The addition of both TFIIA and TFIIB to the TBP–DNA complex further increased the high FRET population at medium forces (72% high FRET at 3.3 pN). However, additive stabilisation of TFIIA and TFIIB was not observed at higher forces because the relative fraction of the high FRET population remained stable even when TFIIA was added to the TBP/TFIIB/DNA complex (high FRET fraction at 6.0 pN: 50% in the absence of TFIIA, 53%

in the presence TFIIA). This suggests that TFIIB represents the crucial factor in the RNAP II system that stabilises the TBP–DNA interaction at higher forces. From a biophysical perspective, it is expected that the force dependence of a binding equilibrium is quantitatively related to the underlying conformational change along the force direction[65] (Supplementary Fig. 6). This can explain why the strong conformational change induced by TBP exhibits a stronger force dependence than, for example, the binding of TFIIB to the TBP/AdMLP DNA complex that goes along with a smaller conformational change (i.e. the FRET shifts indicate a distance change of the dyes of 0.3 nm for TFIIB binding in contrast to 2.8 nm for TBP binding). A more quantitative analysis of the force dependence of individual binding equilibria is not possible as we cannot resolve all the populations in the experiments involving more than one protein.

The addition of the TFIIB-like factor Brf2 to the U6/TBP complex also resulted in a stabilisation of the TBP/DNA complex

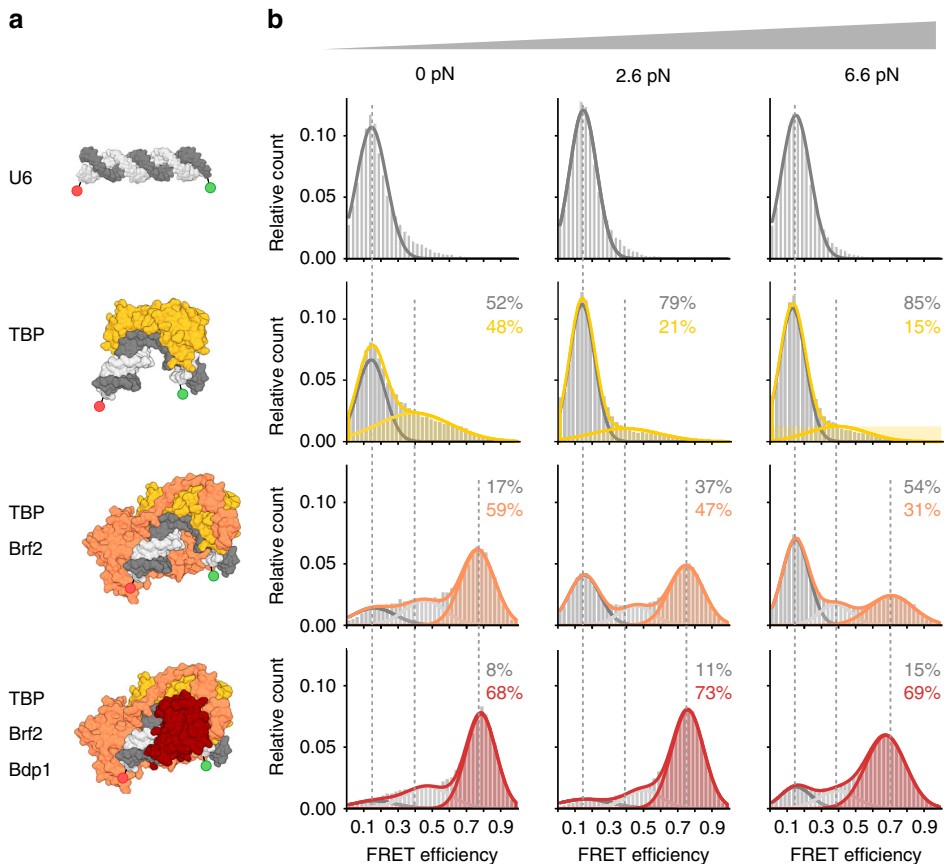

**Fig. 3 Force dependency of promoter binding of RNAP III initiation factors at a canonical RNAP III promoter. a** Structural model (PDB: 5N9G) of the U6 snRNA promoter (U6, dark grey) in an unbent conformation and the bent state bound by TBP (yellow), TBP/Brf2 (orange) and TBP/Brf2/Bdp1 (red). **b** Single-molecule FRET measurements on diffusing molecules monitor TBP-induced DNA bending after the addition of TBP (20 nM), TBP/Brf2 (20 nM) or TBP/Brf2/Bdp1 (20 nM) to the U6 DNA origami force clamps at increasing forces (0, 2.6 and 6.6 pN). FRET efficiency histograms showing the relative distribution between the unbent DNA state (low FRET state, $E = 0.19$, grey) and TBP-induced bent states in the absence and presence of additional initiation factors (high FRET population, $E = 0.39$ (TBP only, yellow), $E = 0.75$ (TBP/Brf2, orange), $E = 0.76$ (TBP/Brf2/Bdp1, red)). Low and high FRET populations were fitted with a Gaussian distribution. Each measurement was carried out at least three times. See also Supplementary Fig. 5 and Supplementary Table 2.

and a shift of the bent DNA population to a higher FRET efficiency ($E = 0.74$) (Fig. 3). In both cases, however, the complex was still force sensitive and only a small fraction (Brf2 31%, TFIIB 30%) of molecules was found in the bent state at 6.6 pN (Figs. 2 and 3, see also Supplementary Fig. 5 for comparison). Interestingly, the addition of TFIIB to the TBP/U6 DNA complex also confers a stabilising effect at low to medium forces (up to 2.6 pN, see Supplementary Fig. 7). However, a stabilising effect of TFIIB at the U6 promoter could only be achieved at comparably high TFIIB concentrations (TFIIB: 200 nM versus Brf2: 20 nM). At higher forces, however, the U6/TBP/TFIIB complex is less stable than the AdMLP/TBP/TFIIB complex. TFIIB binds the TBP/U6 DNA complex via a direct interaction with TBP and a contact to the BRE[66]. As the U6 promoter lacks the BRE, stabilising effects of TFIIB may be reduced. In previous studies, we observed a significant increase in the lifetime of the complexes when Brf2 was added to the TBP/DNA complex[57]. The addition of the Bdp1 SANT domain (residues 241–396, in contrast to the version with residues 130–484 used in this study) to the TBP/Brf2/U6 DNA complex did increase the complex lifetime by 43% when linear promoter DNA was used for smFRET measurements[57]. Hence, we wondered whether Bdp1 influences the complex stability when the DNA experienced increased strain. Probing the force sensitivity of the U6/TBP/Brf2/Bdp1 complex showed that even at 6.6 pN, the majority of molecules (69%) was found in a bent

DNA state. We therefore conclude that in the RNAP III system, Bdp1 is the decisive initiation factor that renders the initiation complex fully stable (Fig. 3). In contrast, TFIIB (even in the presence of TFIIA) cannot reach comparable levels of stabilisation (only 53% of the molecules are found in the bent state) rendering the minimal initiation complex of the RNAP III transcription system more stable than its RNAP II counterpart.

**Increased DNA strain destabilises the TBP–DNA interaction.** Previous measurements showed that the TBP-DNA interaction is dynamic[49,57]. This gave us the opportunity to ask whether the increase in strain reduces the lifetime of the TBP/DNA complex (enhanced TBP dissociation with increase in force) or prolongs the lifetime of the unbent DNA state (inhibited TBP association with increase in force).

To answer this question, we use two different strategies adapted to the underlying kinetics of the association/dissociation process. Slow kinetics in the minutes time regime were measured by acquiring smFRET distributions at different time points after mixing the constituents of the transcription complex using confocal microscopy. Faster kinetics were measured by monitoring the high-FRET and low-FRET state lifetimes directly on single immobilised complexes employing TIRF microscopy. Time-resolved smFRET measurements show that at 0 pN, the AdMLP/TBP complex is an order of magnitude ($\tau_{bent} = 311$ s)

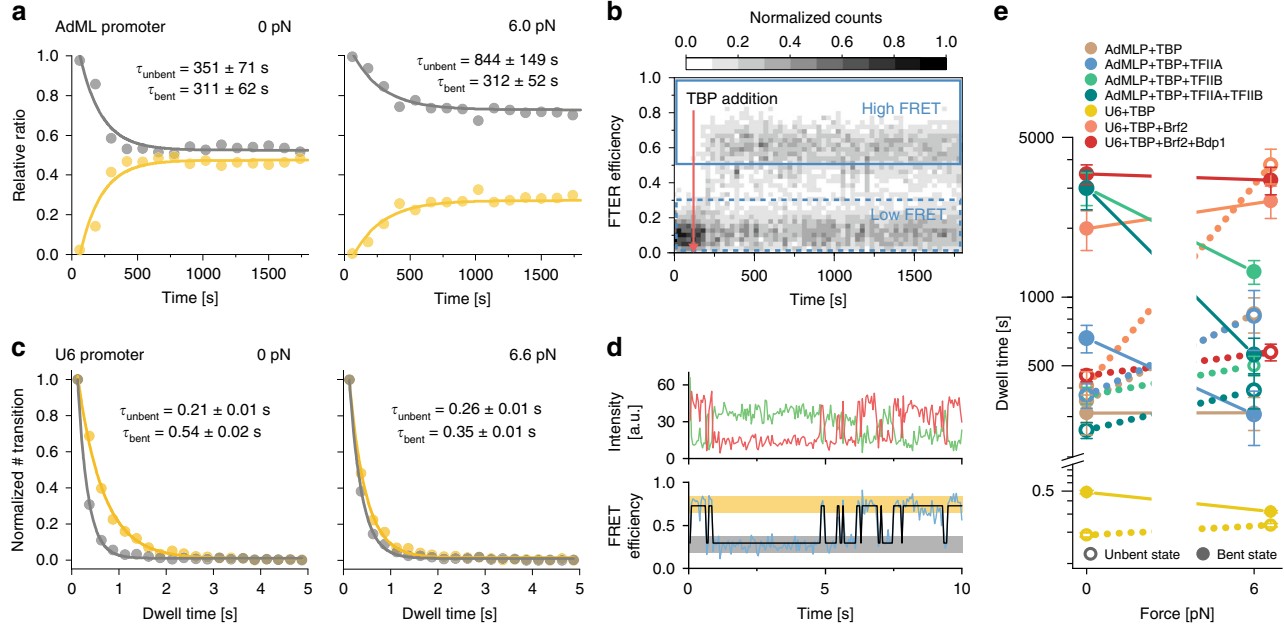

**Fig. 4 Kinetic analysis of the influence of force on TBP-induced DNA bending. a** Relative ratios of low FRET (unbound DNA, grey) to high FRET state (TBP/DNA complex, yellow) in a kinetic experiment performed on diffusing molecules showing the relative fraction of the TBP/DNA complex to the unbent U6 promoter at 0 and 6.0 pN. Dwell times were calculated by deconvolution with a perturbation-relaxation model. Data were fitted with a mono-exponential function. **b** Representative FRET efficiency–time plot of a single confocal time-course experiment at 0 pN force. TBP (20 nM) was added at 2 min (red arrow). Areas used for calculating the ratio of low and high FRET are indicated by blue boxes. **c** Dwell-time histograms of the U6 promoter in the unbent (grey) and bent (yellow) state at 0 and 6.6 pN force. **d** Representative donor (green) and acceptor (red) intensity–time trace from single-molecule TIRF microscopy experiments and the resulting FRET efficiency (blue) fitted with the idealised two-state trace (black) of TBP binding to the U6 promoter at 0 pN force. The low FRET and high FRET states are highlighted in grey and yellow, respectively. **e** Comparison of dwell times in the bent and unbent state for TBP-containing initiation complexes at 0 and 6.0 pN (AdMLP) or 6.6 pN (U6). Mean ± s.e.m. derived via error propagation from the exponential fit (see Supplementary Table 3 for the number of analysed molecules). Connecting lines between data points are meant as visual guides and do not represent interpolations (Supplementary Figs. 8 and 9 and Supplementary Table 3).

more stable than the U6/TBP complex ($\tau_{bent} = 0.54$ s). This is in agreement with previous observations using linear dsDNAs[57]. Increased force does not affect the bent state lifetime of AdMLP/TBP complexes ($\tau_{bent}$: 311 vs. 312 s), whereas the U6/TBP interaction is slightly reduced $\tau_{bent}$: 0.54 vs. 0.35 s). In contrast, increased force leads to an increase of the unbent state lifetime for the AdMLP/TBP complex ($\tau_{unbent}$: 351 vs. 844 s), whereas the U6–TBP complex is mostly unaffected ($\tau_{unbent}$: 0.21 vs. 0.24 s) (Fig. 4, Supplementary Fig. 8 and Supplementary Table 3). These data suggest that two factors contribute to the reduction of the bent DNA states at higher forces: (i) destabilisation of the TBP/DNA complex with increased probability of TBP dissociation from the DNA at higher forces (spring-loaded TBP ejection mechanism) in case of the U6 promoter and (ii) a decreased probability of the TBP to form a stable complex with DNA (TBP entry denial) at the AdMLP. It seems plausible that DNA under strain does provide less flexibility between the bases for the two phenylalanines pairs to insert into the DNA and thereby entry of TBP into the DNA is denied. The interaction of the already inserted phenylalanines with the bases of the DNA, on the other hand, may be reduced at higher DNA strain, leading to ejection at high strains. Kinetic measurements were also performed with the TBP/TFIIA, TBP/TFIIB and TBP/TFIIA/TFIIB complex in interaction with the AdMLP and TBP/Brf2, as well as TBP/Brf2/Bdp1 in interaction with the U6 promoter, respectively (Supplementary Fig. 9). It is important to note, however, that the kinetic experiments did not allow us to disentangle subpopulations and to differentiate between sub-assemblies of the initiation complexes when more than one factor was added to the sample. Hence, kinetic experiments performed with multiple factors

added are providing a general picture about the stabilising contributions of the factors used. These data show that the stabilisation by additional transcription factors prolong the lifetime of the bent state, while the lifetime of the unbent state is only marginally influenced. For example, at 0 pN addition of TFIIA increases the lifetime of the bent state by a factor of two ($\tau_{bent} = 661$ s) and addition of TFIIB by a factor of ten ($\tau_{bent} = 3009$ s). Congruent with the distribution of FRET states, no additive effect of TFIIB and TFIIA is observed as the lifetime of the bent state ($\tau_{bent} = 3000$ s) is nearly identical to the lifetime of the bent state when adding TFIIB only. Increasing the force to 6 pN leads to reduced lifetimes in the bent state. Even in the presence of TFIIB and TFIIA, the lifetime of the bent state is reduced to 561 s. Similar observations were made for the U6 promoter: addition of Brf2 and Bdp1 increases the lifetimes of the bent state. However, an additive effect of Brf2 and Bdp1 stabilisation could be monitored (Brf2: $\tau_{bent} = 1994$ s versus Brf2/Bdp1: $\tau_{bent} = 3457$ s). Interestingly, lifetimes of the AdMLP/TBP/TFIIBTFIIA and the U6/TBP/Brf2/Bdp1 promoter complexes are very similar at 0 pN. However, at higher forces, the stabilisation conferred by Brf2/Bdp1 is more force resistant as almost no change in the lifetime of the bent state was observed ($\tau_{bent} = 3253$ s) (Fig. 4E and Supplementary Fig. 9).

## Discussion

During the initiation phase of transcription, the transcriptional machinery is assembled at the promoter. The minimal factor requirement for transcription initiation consists of TBP and TFIIB to recruit RNAP II and TBP, Brf1 or Brf2 and Bdp1 and additionally SNAPc to recruit RNAP III. One of the interesting

questions in this context is why the RNAP III machinery relies on a third basal initiation factor not conserved in the RNAP I or RNAP II system? Based on our data, part of the answer might be found in the fact that promoter DNA—rather than being a rigid stick-like molecule—is part of a complex chromatin super-structure with dynamic structural variability and consequently subject to mechanical forces in the dynamic landscape of chromatin that is constantly exposed to changes by chromatin remodellers and gene activators[60]. This also includes loop formation and tight nucleosomal packaging that exerts mechanical forces on the DNA[67,68]. Additionally, attractive interaction between nucleosomes mediated by the histone tail domains have recently been observed using DNA nanotechnology[61]. These close-range interactions vary in strength between −0.3 and −8 kcal/mol, which falls into the range covered by our experiments (1 kcal/mol = 4.18 pN·nm)[61,69–72] (Supplementary Fig. 10). However, the chromatin landscape and consequently the forces that act on the promoter DNA differ between RNAP II and III promoters. In this work, we investigated the force sensitivity of transcription initiation factor assembly at the promoter DNA at variable forces employing a method to carry out force measurements based on a DNA origami force clamp[63]. Combined with a smFRET assay, we were able to quantify TBP-induced promoter DNA bending and to evaluate the influence of additional initiation factors.

Using identical TBP concentrations, we found that human TBP bends the U6 snRNA promoter less efficiently under force than the AdMLP. The difference in binding and bending behaviour might be due to the difference in base composition: only four out of eight bases of the TATA sequence of the U6 promoter sequence match the human consensus TATA-box sequence[8]. In contrast, the AdMLP provides a perfect TATA box. This is also reflected in the bent/unbent state lifetime measured for both complexes (Fig. 4). Here, mainly the unbent state lifetime increases with force, thus the AdMLP DNA/TBP complex with its higher lifetime is less effected than the transient U6 DNA/TBP complex. Our data show that TBP in conjunction with TFIIA forms stable but force-sensitive complexes, while the addition of TFIIB leads to force-resistant complexes at the prototypical RNAP II AdMLP. The long lifetime of the TBP/DNA complex, the observed stabilising effect of TFIIA and TFIIB and the increase in bending angle upon addition of TFIIB is consistent with previous smFRET measurements using yeast TBP/TFIIB[30,49]. In the RNAP III transcription system, we observed that the TFIIB-like factor Brf2 also enhances the stability of the TBP/DNA complex[57]. Interestingly, the addition of the third initiation factor, Bdp1, yields an outstandingly stable initiation complex at the U6 promoter. It is noteworthy that the spliceosomal U6 RNA and other RNAP III gene products are highly expressed. This in turn requires robust formation of initiation complexes at the promoter as transcriptional regulation cannot take place at the level of elongation at these extremely short genes. Hence, the RNAP III-exclusive initiation factor Bdp1 plays a decisive role in transcription initiation as it allows the maintenance of fully assembled TFIIIB-promoter DNA complex. The stable anchoring of initiation proteins as well as the RNAP III is furthermore of crucial importance as RNAP III is thought to undergo extensive cycles of facilitated re-initiation[73–75]. RNAP III only transcribes very short RNAs (5S rRNA, tRNAs, U6 snRNA) and biochemical and recent structural data suggest that RNAP III, in contrast to RNAP II, might not disengage from the promoter during transcription elongation, but possibly remains bound to the promoter and re-initiates directly after termination[73–76]. Hence, initiation factors at the promoter are situated at a DNA section that is topological restrained on the one hand side by the −1 nucleosome, which is stably positioned at −150 bp[76] and a

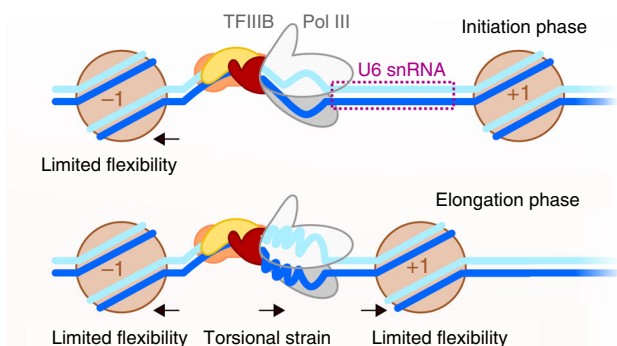

**Fig. 5 Model describing the role of DNA strain in RNA polymerase III initiation complexes.** On the U6 snRNA promoter, the −1 nucleosome is firmly positioned close to the upstream promoter region, limiting DNA flexibility. Continuous transcription of the U6 snRNA by promoter-bound RNA polymerase III creates torsional strain. The +1 nucleosome is positioned downstream of the gene body. Proteins and nucleic acids are colour coded as follows: TBP (yellow), Brf2 (orange), Bdp1 (red), RNA polymerase II/III (grey), nucleosome (brown), template strand (blue) and non-template strand (cyan).

firmly associated transcribing RNAP. Upon promoter opening of the DNA by RNAP III in concert with Bdp1, the DNA section experiences torsional strain as the DNA is unwound and the strain cannot be released due to the static nucleosome and RNAP III that represent fixed boundaries (Fig. 5). Hence, TFIIIB is likely to experience mechanical forces that are compensated by the extremely stable initiation complex. Moreover, in a model where the polymerase remains bound to the promoter, strain would build up during transcription between the promoter binding site and the active site due to the increasing amount of transcribed DNA that has to be accommodated in the polymerase. This additionally increases the forces that the transcription initiation complex has to withstand.

The situation is different at RNAP II promoters as RNAP II transcribes mRNAs that can be hundreds of base pairs in length and re-initiation does not seem to play a role. Another point to consider is that RNAP II and III promoters display a nucleosome-depleted region around the TSS, but a conserved +1 nucleosome is found at position +40 in genes with elongating RNAP and +10 in silent genes (RNAP II)[77,78] and 220 bp (RNAP III)[76]. As the position of the +1 nucleosome does not show a strong sequence dependency and its position appears to be flexible when nucleosomes are reconstituted on naked DNA in vitro[78], it has been speculated that initiation factors situated at the promoter help to establish the position of the +1 nucleosome[60,79]. This might be especially relevant for RNAP II genes where the +1 nucleosome is found in close proximity to the TSS. In this case, initiation factors need to be stably attached at the promoter in order to avoid displacement by the nucleosome. Whereas TFIIA increases the stability of the TBP/DNA complex, it does not confer force resistance. Instead, TFIIB acts as the initial stabilising factor at RNAP II promoters to secure TBP at the DNA and this minimal initiation complex can be further extended by additional initiation factors and ultimately extended to include the Mediator complex[18]. Homologue factors are not found in the RNAP III system, but our studies show that the addition of Bdp1 to the RNAP III initiation complex is necessary to maintain an active initiation complex even when the transcribing RNAP III potentially causes increased DNA strain in the promoter DNA. Structural studies show that Brf2, the Bdp1 SANT domain and the Bdp1 linker completely envelop the DNA (Supplementary Fig. 11). Even though TFIIB and TFIIA occupy a comparable location as Brf2 and Bdp1 relative to TBP on the DNA, the

structural element of the Bdp1 linker is missing from TFIIA or TFIIB in the RNAP II system, yielding a comparably less stable pre-initiation complex (Supplementary Fig. 11).

This indicates that the tension on the DNA might be a mechanism of gene regulation. The packaging, histone placement, action of the replication machinery and binding of regulatory proteins will certainly have an impact on the tension that the initiation complex is exposed to. Thus, besides steric effects, tension influences transcription. On the other hand, after the transcription initiation complex has formed (i.e. more than one transcription factor is assembled at the promoter), the lifetime of the complex becomes independent of force. This might indicate that after the decision for transcription was taken, the process should become independent of mechanical factors ensuring that the RNAP enters the elongation phase of transcription.

## Methods

**Protein expression and purification.** All proteins were expressed and purified as described previously[57,80]. Briefly, human TBP (169–339, N-terminal His-Tag), Brf2 (62–419, N-terminal His-tag), Bdp1 (130–484, C-terminal His-tag[20,57]) and TFIIB (full length, C-terminal His-tag) were in expressed in *Escherichia coli* Rosetta (DE3) pLysS (TBP, Brf2, TFIIB) or BL21-CodonPlus (DE3)-RIL (Bdp1). Cells were lysed in lysis buffer (750 mM NaCl, 10 mM imidazole, 50 mM HEPES pH 7.9, 10% glycerol, 5 mM 2-mercaptoethanol, protease inhibitors tablets (Pierce) and DNaseI) and purified using HisTrap and heparin columns, followed by size-exclusion chromatography (Superdex 200 16/600 column) in a buffer containing 50 mM HEPES pH 7.9, 500 mM ammonium acetate, 10% glycerol and 1 mM TCEP.

Human TFIIA (single-chain variant, where subunits α, β and γ are connected via linkers[80]) was expressed in *E. coli* BL21 (DE3) cells at 18 °C overnight. Cells were lysed in binding buffer (20 mM Tris pH 7.4, 150 mM NaCl and complete protease inhibitor tablet (Pierce) using a French press and subsequently cleared by centrifugation at 40,000 r.c.f. for 45 min. The protein was then purified using Talon affinity chromatography and eluted in elution buffer (20 mM Tris pH 7.4, 150 mM NaCl and 250 mM imidazole). Eluted protein was dialysed in 20 mM Tris pH 7.4, 150 mM NaCl, 0.5 mM EDTA and 1 mM dithiothreitol (DTT), and further purified using a heparin column. Eluted protein was further subjected to size-exclusion chromatography (Superdex 75 column) equilibrated in SEC buffer (20 mM Tris pH 7.4, 150 mM NaCl, 0.5 mM EDTA and 1 mM DTT).

**Cloning of promoter DNA sequences into the M13 DNA origami scaffold.** The Force-clamp origami used in this work is based on the M13mp18 ssDNA. The multiple cloning site of the ssDNA phage DNA is located within the spring region of the force clamp, and the two different RNAP promoters were cloned between the *Bam*HI and *Hin*dIII restriction sites of the multiple cloning site. The AdMLP and U6 promoters were assembled by means of hybridisation of 5'-phosphorylated forward and reverse oligonucleotides (Supplementary Table 1). Annealing of the forward and reverse oligonucleotides generate *Bam*HI and *Hin*dIII sticky ends. Cloning was performed using the replicative form (dsDNA) of the M13 phage.

**Preparation of M13 phage ssDNA.** Phage ssDNA production was carried out in *E. coli* XL1blue cells. Two hundred and fifty millilitres of LB phage medium (low salt LB broth, 5 g/L NaCl) supplemented with 5 µg/mL tetracycline was inoculated with 250 µL of an overnight grown XL1blue cell culture. The culture was incubated at 37 °C with vigorous shaking until it reached an optical density of 0.5, then it was inoculated with 100 µL of M13 phage supernatant and incubated for another 5 h. The cells were pelleted by centrifugation (15 min at 6000 r.c.f.), and the supernatant clarified via centrifugation.

For phage ssDNA purification, 10 g of polyethylene glycol (PEG) 8000 and 7.5 g of NaCl were added to the supernatant and stirred for 30 min at room temperature (RT). After that, the solution was centrifuged at 5000 r.c.f., 4 °C for 30 min, the supernatant discarded and the pellet resuspended in 2.5 mL of TE buffer. The phage suspension in TE buffer was centrifuged again at 16,000 r.c.f., 4 °C for 10 min and the supernatant processed for ssDNA extraction. Five millilitres of PPB2 lysis buffer (0.2 M NaOH, 1% SDS) was added to the supernatant, and after 3 min incubation, it was neutralised with 3.75 mL of PPB3 (3 M KOAc titrated to pH 5.5 with glacial acetic acid). The mixture was centrifuged twice at 16,000 r.c.f., 4 °C, 15 min and the pellet discarded. Twenty millilitres of ice cold 100% ethanol was used to precipitate the DNA. Afterwards, the solution was centrifuged at 16,000 r.c.f., 4 °C, 30 min, the supernatant removed and the pellet was air dried. Finally, the ssDNA pellet was dissolved in 1 mL TE buffer.

**Preparation of doubly labelled ssDNAs.** Doubly labelled ssDNAs were prepared from individual DNA strands that carry either the donor or the acceptor fluorophore (Supplementary Table 1). The final DNA strand carries both dyes and is complementary to the promoter region of the origami scaffold. Ten micromolar of the appropriate donor strand (_D), acceptor strand (_A) and complementary

**Table 1 Temperature ramp for the folding of DNA origami force clamps.**

| Temperature (°C) | Time per °C (min) | Temperature (°C) | Time per °C (min) |
|---|---|---|---|
| 65 | 2 | 44 | 75 |
| 64–61 | 3 | 43 | 60 |
| 60–59 | 15 | 42 | 45 |
| 58 | 30 | 41–39 | 30 |
| 57 | 45 | 38–37 | 15 |
| 56 | 60 | 36–30 | 8 |
| 55 | 75 | 29–25 | 2 |
| 54–45 | 90 | 8 | Storage |

ligation strand (_Lig) were hybridised in 100 µL annealing buffer (Tris HCl pH 8.0, 150 mM NaCl), heated to 90 °C for 3 min and cooled down to 20 °C for 2 h. For the ligation, 20 µL 10× T4 ligase buffer (NEB), 70 µL Millipore water and 10 µL T4 DNA ligase (NEB) were added to the hybridisation reaction and incubated for 60 min at 20 °C.

To purify the ligated ssDNA, the DNA was separated on a preparative denaturing TBE gel (15% (v/v) acrylamide/bis-acrylamide (19:1), 6 M urea). To this end, RNA loading buffer (47.5% glycerol (v/v) 0.1% (v/v) SDS, 0.5 mM EDTA) was added to the ligation reaction and the sample was heated to 80 °C and cooled on ice. The DNA was separated at 200 V over 40 min. The gel was visualised under UV light and the band corresponding to the doubly labelled DNA strand was excised and pulverised. DNA was extracted by adding 1 mL of 1× TBE buffer and shaking at 4 °C for 2 h. The gel debris was pelleted via centrifugation at 15,000 r.c.f. for 30 min (repeated once). The DNA was precipitated by the addition of 1/10th volume of ammonium acetate solution (3 M, pH 5) and 2.5 volumes of ethanol. The sample was incubated at −80 °C for 1 h, followed by a centrifugation step for 1 h at 4 °C. The supernatant was carefully decanted and the DNA was washed by the addition of 5 mL of 70% ethanol and 30 min centrifugation at 15,000 r.c.f. The supernatant was completely removed, the pellet dried for 10 min at 20 °C and resuspended in 10 mM Tris HCl pH 8.0 + 50 mM NaCl.

**DNA origami preparation and purification.** DNA origamis were assembled as described previously[63] (see Supplementary Note 1). In brief, scaffold DNA (25 nM), core staple strands (200 nM), force staple strands (400 nM), biotin adapter staple strands (200 nM) and the complementary doubly labelled promoter DNA strand (200 nM) were mixed in folding buffer (10 mM Tris pH 7.6, 1 mM EDTA, 20 mM MgCl$_2$, 5 mM NaCl) and subjected to a multistep thermocycler protocol (Table 1). Afterwards, the origami was purified by the addition of one volume of 2× pre-cipitation puffer (Tris HCl pH 7.6, 1 mM EDTA, 500 mM NaCl, 15% (w/v) PEG-8000) and centrifugation at 20,000 r.c.f. for 30 min at 4 °C. Afterwards, the super-natant was decanted and the pellet resuspended in 30 µL folding buffer for 30 min at 30 °C under constant shaking. All purification steps were repeated once.

**Restriction digestion of origami scaffolds.** In order to generate force clamps with 0 pN force, the spring strand was cleaved with a *Bam*HI restriction endonuclease. To this end, 200 µM of the scaffold DNA and 3× molar excess of *Bam*HI_comp strand were hybridised in FastDigest Green buffer (Thermo Scientific) by heating the sample to 90 °C, followed by gradual cooling to 20 °C over 2 h. Afterwards, 1 U of FastDigest *Bam*HI (Thermo Scientific) was added, incubated at 37 °C for 4 h. Subsequently, *Bam*HI was heat inactivated at 80 °C for 10 min.

**Sample preparation and transmission electron microscopy imaging of DNA origami force clamps.** Electron microscopy grids (copper, 400 mesh; PlanoEM, Germany) were pre-coated with a ~8-nm-thick carbon support film generated with a "Turbo Carbon Coater" (Cressington Scientific Instruments, UK). Four micro-litres of force clamp solution (10 nM) was applied to glow-discharged grids for 30 s, blotted off and the grids washed 3× with water (bi-distilled water). Grids were dried and imaged without staining using a JEM2100F transmission electron microscope (Jeol, Japan) equipped with TemCamF416 detector (TVIPS, Germany). A total of 94 micrographs were semi-automatically collected at 40,000 magnification (2.7 Å/pix) and a defocus range of −2.5 to −4.5 µm using the Serial-EM software package[81]. Contrast transfer function (CTF) estimation and manual particle picking were carried out in Relion 3[82]. A total of 2133 particles (2× binned; box size 180 pixels/ 100 nm) were extracted, contrast inverted and 2-D classified into three classes with CTF-amplitude correction from the first peak onward.

**Surface preparation.** Silica microscope slides used for TIRF experiments were prepared as described before[57]. Briefly, fused silica slides (Plano) were cleaned in peroxomosulfuric acid (70% (v/v) sulfuric acid; Fisher Scientific, 10% (v/v) hydrogen peroxide; Sigma-Aldrich) for 30 min and washed with Millipore water under sonication. Afterwards, the slides were incubated in methanol for 20 min and

sonicated for 5 min. For silane passivation, the slides were incubated in a freshly prepared N-[3-(Trimethoxysilyl)-propyl]ethyldiamine (Sigma-Aldrich) solution (2% (v/v) in methanol with 4% (v/v) acetic acid) for 20 min, rinsed with methanol five times and an additional 20 times with Millipore water. The slides were dried for 1 h at 37 °C. For PEG passivation, 100 μL of freshly prepared passivation solution (200 mg/mL methoxy-PEG succinimidyl valerate 5000 (Laysan Bio), 5 mg/mL biotin-PEG (Laysan Bio) in 1 mM NaHCO₃) was sandwiched between a slide and a coverslip, incubated for 2 h and rinsed with Millipore water 20 times. The slides and coverslips were fully dried at 37 °C, vacuum sealed in plastic tubes and stored at −20 °C.

**TIRF immobilisation assay.** Single-molecule FRET measurements on immobilised DNA/protein complexes were carried out in custom-built flow chambers based on fused silica slides passivated with PEG. Flow chambers were prepared and assembled as described before[49].

For fluorescence measurements, the flow chamber was incubated with 0.1 mg/mL NeutrAvidin (Pierce) in 1× TBS (125 mM Tris/HCl pH 8, 150 mM NaCl) for 5 min and washed with 500 μL T78 buffer (100 mM Tris/HCl pH 7.8, 60 mM KCl, 5 mM MgCl₂, 0.5 mg/mL bovine serum albumin (BSA), 1% (v/v) glycerol). Afterwards, the chamber was flushed with DNA origami force clamps (10 pM in folding buffer) for 5 s and washed with 500 μL T78 buffer. The chamber wash flushed with photo stabiliser buffer (T78 buffer with 2 mM Trolox, 1% (w/v) D-glucose, 7.5 U/mL glucose oxidase type VII (Sigma-Aldrich), 1 kU/mL catalase (Sigma-Aldrich)) supplemented with 10 nM human TBP and incubated for 5 min before starting video acquisition.

**Wide-field single-molecule detection and data analysis.** Time-resolved single-molecule fluorescence measurements were performed on a homebuilt prism-type total internal reflection setup based on a Leica DMi8 inverse research microscope. Fluorophores were exited with a 532-nm solid-state laser (Coherent OBIS) with a power of 30 mW and 637 nm diode laser (Coherent OBIS, clean-up filter ZET 635/10, AHF Göttingen) with a power of 50 mW employing alternating laser excitation (Multistream, Cairn Research, UK)[83]. The fluorescence was collected by a Leica HC PL Apo ×63 NA 1.20 water-immersion objective and split by wavelength with a dichroic mirror (HC BS 640, AHF) into two detection channels that were further filtered with a 582/75 bandpass filter (Brightline HC, AHF) in the green channel and a 635-nm long-pass filter (LP Edge Basic, AHF) in the red detection channel. Both detection channels were recorded by one EMCCD camera (Andor IXon Ultra 897, EM-gain 20, framerate 40 Hz, 400 frames) in a dual-view configuration (Optosplit III, Cairn Research).

The videos were analysed employing the iSMS software package[84] using the programs defaults settings. Molecule spots were detected using a threshold of 100 for ATTO 532 and ATTO 647N spots. FRET efficiencies were calculated as proximity ratios from fluorescence intensity–time traces that were corrected for background fluorescence using the average intensity of all pixels with a 2 pixel distance to the molecule spot. The FRET efficiency $E$ and the stoichiometry factor $S$ describing the relative ratio between donor and acceptor signal for each molecule were calculated as:

$$E = \frac{I_{AD} - \alpha \times I_{DD} - \delta \times I_{AA}}{\gamma \times I_{DD} + (I_{AD} - \alpha \times I_{DD} - \delta \times I_{AA})}, \quad (1)$$

$$S = \frac{\gamma \times I_{DD} + (I_{AD} - \alpha \times I_{DD} - \delta \times I_{AA})}{\gamma \times I_{DD} + (I_{AD} - \alpha \times I_{DD} - \delta \times I_{AA}) + I_{AA}}, \quad (2)$$

where $I_{DD}$ and $I_{AD}$ are the donor intensity and acceptor intensity upon donor excitation, respectively. The correction factors $\gamma$, $\alpha$ and $\delta$ were calculated from individual time traces as:

$$\alpha = \frac{I_{DA,t_2}}{I_{DD,t_2}}, \quad (3)$$

$$\delta = \frac{I_{DA,t_2}}{I_{AA,t_2}}, \quad (4)$$

$$\gamma = \frac{I_{DA,t_2} - I_{DA,t_1}}{I_{DD,t_2} - I_{DD,t_1}}, \quad (5)$$

where $t_1$ is the time interval before and $t_2$ the time interval after photobleaching of the fluorophore.

For TBP dwell-time histograms, traces showing dynamic switching between FRET states were fitted with the vbFRET algorithm[85] limited to two states (Supplementary Figure 8). FRET efficiency histograms were calculated from all frames of traces showing dynamic switching between states with a stoichiometry value between 0.4 and 0.6 and were fitted with a Gaussian distribution. All states calculated with vbFRET with a FRET efficiency within the full-width at half-maximum of a fitted FRET population were used to calculate the dwell-time histogram. The histograms of at least three independent experiments were normalised and fitted with a mono-exponential decay function to calculate the mean dwell time in the high FRET state (TBP bound to DNA).

**Confocal single-pair FRET measurements.** Prior to sample loading, the sample chambers (Cellview slide, Greiner Bio-One) were passivated with 10 mM Tris/HCl pH 8 with 2 mg/mL BSA for 10 min and washed once with T78 buffer.

For equilibrium measurements (Figs. 2 and 3 and Supplementary Figs. 4, 5 and 7) complexes were formed with 20 pM DNA origami and 20 nM TBP, Brf2 and Bdp1 or 200 nM TFIIB and incubated for 30 min at RT in T78 buffer with 2 mM DTT.

For time-course experiments (Fig. 4 and Supplementary Fig. 9), 20 pM DNA origami and 20 nM Brf2 and Bdp1 or 200 nM of TFIIB in T78 buffer with 2 mM DTT were added to the sample chamber and data acquisition was started to measure the unbound DNA state. After 2 min, TBP was added to initiate complex formation.

Single-molecule fluorescence of diffusing complexes was detected with a MicroTime 200 confocal microscope (PicoQuant) equipped with pulsed laser diodes (532 nm: LDH-P-FA-530B; 636 nm: LDH-D-C-640; PicoQuant/clean-up filter: ZET 635; Chroma). The fluorophores were excited at 20 μW using pulsed interleaved excitation. Emitted fluorescence was collected using a 1.2 NA, ×60 microscope objective (UplanSApo ×60/1.20 W; Olympus) and a 50-μm confocal pinhole. A dichroic mirror (T635lpxr; Chroma) separated donor and acceptor fluorescence. Additional bandpass filters (donor: ff01-582/64; Chroma; acceptor: H690/70; Chroma) completed spectral separation of the sample fluorescence. Each filtered photon stream was detected by an individual APD (SPCM-AQRH-14-TR, Excelitas Technologies) and analysed by a TCSPC capable PicoQuant HydraHarp 400.

**Data analysis.** Data analysis of confocal FRET measurements was performed with the software package PAM[86]. Photon bursts of diffusing molecules were determined by an all-photon burst search (APBS, parameters: $L = 50$, $M = 20$, and $T = 500$ μs) and an additional dual-channel burst search (DCBS, parameters: $L = 50$, $M_{GG+GR} = 20$, $M_{RR} = 20$, and $T = 500$ μs). Burst data were corrected for donor leakage and direct excitation of the acceptor (determined from APBS according to ref. [87]) as well as $\gamma$ and $\beta$ (determined from DCBS ES histograms using an internal fit on multiple E/S separated FRET populations). The data were binned (bin size = 0.025) and the mean value for the triplicate of experiments was plotted as E histogram (see Supplementary Figure 5c for mean histograms with standard deviation). The histograms were fitted with a single (DNA) or triple Gaussian fit due to a medium FRET density ($E = 0.4$–$0.5$) that we observed in all origami experiments involving proteins, but not in control measurements with linear promoter constructs. Therefore, we assume that this effect has no connection to the biological system. To account for this density during data analysis, we added an additional Gaussian with a fixed area to all our fits to improve the overall fit quality (see Supplementary Table 2), but did not include it in the Results and Discussion sections.

**Kinetics measurements.** Data were processed as above. All bursts were sorted according to their FRET efficiency (low FRET for $E < 0.3$ and high FRET for $E > 0.6$) and binned by macrotime (bin size = 2 min). Low FRET and high FRET bins were normalised to the combined sum to determine relative ratios of both populations, which were plotted against time and fitted with a mono-exponential function. The fit-derived decay constant and $y$-offset ($y0$, equivalent to low FRET ratio at equilibrium) for the low FRET population were used to determine dwell times in the high FRET and low FRET state via deconvolution with a perturbation-relaxation model (see Supplementary Information).

**Confocal kinetics measurements.** The unbent and bent state dwell times for the highly stable AdMLP/TBP/(TFIIB) and U6/TBP/Brf2/(Bdp1) complexes were measured via confocal single-molecule experiments in solution. All factors except TBP were mixed prior to data acquisition. TBP was added 2 min after the start and the decay of the low FRET and increase of the high FRET population was monitored over time. Fitting this data to a mono-exponential model yields the decay rate $\tau$ and the unbent fraction $[u]_{new}$ in equilibrium with all proteins (Supplementary Table 3). This decay constant translates to the complex assembly rate that is superimposed with the simultaneous complex disassembly. We used a perturbation-relaxation kinetics model to extract both rates from the data.

The DNA–TBP system is in an equilibrium of two states, the unbent state and the bent state,

$$\text{unbent} \underset{k_{bent} = \tau_{unbent}}{\overset{k_{bent} = \tau_{unbent}}{\rightleftarrows}} \text{bent}, \quad (6)$$

where $k_{bent}$ is the bending rate and $k_{unbent}$ is the unbending rate. Conversely, $\tau_{unbent}$ and $\tau_{bent}$ are the average lifetimes of a molecule in the unbent or bent state. The equilibrium constant $K$ is given by the ratio of the rates:

$$K = \frac{k_{bent}}{k_{unbent}}. \quad (7)$$

The system is perturbed by a change in the TBP concentration and will relax in a first-order process to its new equilibrium. Since $k_{bent}$ is concentration dependent,

the unbent fraction $[u]$ will vary according to:

$$[u](t) = [u]_{new} + Ae^{-\frac{t}{\tau}}. \quad (8)$$

The amplitude $A$ is the change of the unbent fraction and the new equilibrium unbent fraction is $[u]_{new}$ The relaxation time $\tau$ is given by both rate constants

$$\tau = \frac{1}{k_{bent} + k_{unbent}}, \quad (9)$$

with the new equilibrium constant and the decay time both rate constants can be calculated for the new equilibrium as:

$$k_{bent} = \frac{K}{\tau(K+1)} = \frac{1}{\tau_{unbent}}, \quad (10)$$

$$k_{unbent} = \frac{1}{\tau(K+1)} = \frac{1}{\tau_{bent}}. \quad (11)$$

**Fluorescence correlation spectroscopy**. Fluorescence correlation spectroscopy analysis was performed with the PicoQuant software package SymPhoTime 64. The acceptor signal of an arbitrary 10 min interval of each data set was autocorrelated according to Eq. (12) and fitted following Eq. (13) with one (dsDNA) or two (origami) diffusion components ($n_{Diff}$). The relative diffusion times given in Supplementary Fig. 3 were calculated with a non-calibrated confocal volume $V_{eff} = 1$. The statistical error shown is the standard error of the fit

$$G(\tau) = \frac{\langle I(t)I(t+\tau) \rangle}{\langle I(t) \rangle^2} - 1, \quad (12)$$

$$G(\tau) = \left[ 1 + T \left[ \exp\left( \frac{\tau}{\tau_{Trip}} \right) - 1 \right] \right] \sum_{i=0}^{n_{Diff}-1} \frac{\rho[i]}{\left[ 1 + \frac{\tau}{\tau_{Diff[i]}} \right] \left[ 1 + \frac{\tau}{\tau_{Diff[i]} \kappa^2} \right]^{0.5}}, \quad (13)$$

where $\tau$ is the correlation time, $I$ is the signal intensity and $t$ is the experiment time, $T$ is the fraction of triplet state molecules, $\tau_{Trip}$ is the lifetime of the triplet state $\rho$ is contribution of the $i$th diffusing species and $\kappa$ is the length to diameter ratio of the confocal volume.

**Reporting summary**. Further information on research design is available in the Nature Research Reporting Summary linked to this article.

## Data availability
The data that support this study are available from the corresponding author upon reasonable request. The underlying Figs. 2 and 3 and Supplementary Figs. 4 and 7 are available as a Source Data file. Source data are provided with this paper.

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

## Acknowledgements

We gratefully acknowledge financial support by the Deutsche Forschungsgemeinschaft (SFB960-TP7 to D.G.). P.T. acknowledges support by the DFG (grant INST 86/1904-1 FUGG), excellence clusters CIPSM (Center for Integrated Protein Science Munich) and NIM (Nanosystems Initiative Munich). T.L. is also supported through NIM and the SFB1032-TP6. F.B.H. and C.E. were supported by DFGs "Emmy-Noether-Programm" [DFG grant no. EN 1204/1-1] and SFB960-TP A8. A.M.V. is supported by a Cancer Research UK Programme Foundation (CR-UK C47547/A21536) and a Wellcome Trust Investigator Award (200818/Z/16/Z). I.B. is a Senior Investigator of the Wellcome Trust (106115/Z/14/Z). A.M.V. has received funding from the European Union's Horizon 2020 research and innovation programme under the Marie Skłodowska-Curie Grant Agreement No. 746635. Furthermore, we would like to thank Dr. Sarah Willkomm for advice on analysing the kinetics data, Michael Pilsl for support on the EM measurements and Elisabeth Piechatschek and Elke Papst for technical assistance.

## Author contributions

D.G. and K.K. conceived the study. K.K. performed the single-molecule measurements. K.K. and T.S. analysed the single-molecule data. J.G., K.G., I.B. and A.V. purified the proteins. T.S., A.M.V., T.L. and P.T. designed and manufactured the DNA origami force clamp. F.B.H. and C.E. carried out electron microscopy measurements and analysed the data. K.K. and D.G. wrote the paper. All authors commented on the paper.

## Competing interests

The authors declare no competing interests.
