## [Peer Review File · Nature Communications]

Reviewers' comments:

Reviewer #1 (Remarks to the Author):

Kramm et al. investigate in this study the stability of the human Pol III preinitiation complex using single-molecule FRET measurements. They challenge the complex using a previously established origami force-clamp technique and compare the obtained results with the Pol II preinitiation complex. The major finding is that the Pol III specific factor Bdp1 increases the formation of the preinitiation complex at elevated forces from 31% to 69%. At low forces Bdp1 does not appear to impact the complex formation. From this the authors speculate that the Bdp1-induced force resistance is important for RNAP III to cope with torsional as well as topological stress from the surrounding chromatin, which is, however, not further tested. I have a bit mixed feelings on this manuscript. Overall the experiments are carefully executed. I am, however, not really competent in judging how important the found Bdp1-induced force resistance on the Pol III preinitiation complex is for the transcription field. Regarding the biophysics of the preinitiation complex formation, I find it a pity that force-dependencies were measured but not further related to known conformational changes of the promoter DNA.

Detailed remarks:

- 1) The second half of the introduction is quite hard to follow since it covers the preinitiation complexes of RNAP II and III for archea, yeast and humans. For non-transcription experts this provides a quite hard time when reading. I suggest to focus the introduction more towards the topic of the study.
- 2) As shown in many previous biophysical investigations, a force-dependence of the equilibrium of a binding reaction can be quantitatively related to an underlying conformational change along the force direction (e.g. early work of the Wuite lab on restriction enzymes). This is barely discussed in the manuscript and also not quantitatively explored. Since the force is applied on the DNA ends, a force dependence would be expected if the distance between the DNA ends is shrinking during the binding step. In my opinion this can explain the force-dependence of all association reactions where the FRET shifts to higher values. In contrast for the additional Bdp1 association no change of the DNA end distance is seen from the FRET measurements, which can explain the absent force dependence. In my opinion this would even suggest that Bdp1 would bind after Brf2?
- 3) The presented kinetic analysis of the preinitiation complex formation is somewhat puzzling to me. From the force dependence of the rates the authors suggest that TBP uses different mechanisms when binding to the AdML compared to the U6 promoter. I have a hard time to understand this given that it is the same protein in both cases such that a similar mechanism would be expected. Can the authors relate the different force dependence of binding and dissociation to DNA conformational changes? Alternatively, can they exclude that the different applied techniques were causing the difference in force dependence? After checking the Supplement I found that the kinetic analysis was also performed for the more complete forms of the preinitiation complexes. This was not mentioned in the main text. Can one draw any helpful conclusion from this additional data?
- 4) There are a few typos in the manuscript, e.g.:
line 62: "angle of 90°C" (:-))
line 69: 2x "to"
line 100: "TFB" should be "TFIIB"?

Reviewer #2 (Remarks to the Author):

Kramm et al use DNA origami-based single-molecule force spectroscopy to analyze occupancies and dwell-times of human Pol II and Pol III transcription factor complexes on their respective

promoters. Their experiments allow them to measure the effect of mechanical force on the occupancy and lifetime of these complexes. The authors show successive stabilization of DNA-bending with increasing numbers of transcription factors. In the case of the Pol II system, using the consensus-TATA-box-containing AdMLP promoter, stable complexes can be obtained by the binding of TBP and TFIIB even under high force. In the case of the Pol III system, using the U6 promoter that features a suboptimal TATA box, binding of the TFIIB-like factor Brf2 stabilizes the bent state less efficiently compared to the TFIIB-AdMLP promoter. However, inclusion of the factor Bdp1 generates a very stable complex even under force.

The authors conclude that in the Pol II system, TFIIB is sufficient for stable anchoring of transcription factors at the promoter, and that in the Pol III system, Bdp1 generates a very stable complex. The authors discuss their findings based on a model in which transcribing Pol III does not leave the promoter, and conclude that the exceptional stability of the TBP-Brf2-Bdp1 complex is required because the complexes experiences high mechanical stress during transcription.

Major Points

1. The conclusion that TFIIB suffices to form stable Pol II promoter complexes is probably not generally correct, but rather depends on the promoter. In fact, in their control experiment in which the U6 promoter was used for measurements with the Pol II factor TFIIB, TFIIB and Brf2 show identical levels of stabilization. However, this experiment is not properly introduced in the text. This experiment also shows that the authors measure the difference in the "bendability" between the U6 and the AdMLP promoter rather than differences between TFIIB and Brf2. The authors should better introduce the experiments and results shown in Suppl. Fig. 5 and discuss them more extensively.
2. The biological novelty of the manuscript is limited. It has been previously demonstrated that Bdp1 confers exceptional stability to the Brf2-Bdp1-TBP-DNA complex. Furthermore, because of the issues listed in point 1) it is difficult drawing general conclusions about the role of TFIIB and Brf2 in forming stable promoter complexes. The novelty therefore lays more in the application of DNA origami-based single-molecule force spectroscopy to compare the force sensitivity of Pol II and Pol III promoter complexes.
3. Recent structural studies have established that the Bdp1 subunit adopts a similar position in the Pol III pre-initiation complex as TFIIA in the Pol II system. In agreement with this, TFIIA has been demonstrated to stabilize the TBP-TFIIB complex on suboptimal promoters. To further increase the impact of the study, it would be interesting to see if addition of TFIIA to the TBP-TFIIB system can confer similar stabilization as Bdp1 does in the Pol III system
4. The interpretation of the results in the discussion section is based on a model in which Pol III stays permanently bound to the promoter during transcription. It should be kept in mind that this remains a hypothetical model. Noteworthy, the cited structure of Pol III bound to TFIIB and an elongation scaffold (Han et al, Cell Discovery, 2018) lacks RNA, indicating that this might be an in vitro artefact through the use of a mismatched DNA bubble. It is also unlikely that the entire transcribed region of a Pol III gene can be accommodated in the active site as stated at page 15, line 373ff. Lastly, up to three Pol III enzymes transcribing the same gene have directly been visualized by electron microscopy in yeast (French et al, MCB 2008), showing that not all transcribing Pol III enzymes are tethered to the promoter. In conclusion, the authors should tone down their statement that Pol III remains always bound to the promoter, but rather state that Pol III might be tethered to the promoter through an interaction with TFIIB during elongation in some cases.

Despite these criticism, this is overall an interesting study using an interesting not yet frequently used technique. Therefore, this study should be published in Nature Communications once the different points listed above have been addressed.

Minor points:

The manuscript contains an amazing (!) number of spelling mistakes. Please check also for consistency of spelling (RNAPII, RNAP II, RNAP vs Pol).

Page 2, lines 32-33: "The TATA-binding protein (TBP) and a transcription factor (TF) IIB-like factor compound the fundamental core of all eukaryotic initiation complexes."

This does not seem to be the case for Pol I were TBP is dispensable in vitro, as also later stated by the authors.

Page 3, line 34: "intiation". Correct spelling

Page 3, line 55: Change "phase" to "phases"

Page 3, line 56: The term "archaeal-eukaryotic RNAP" suggests that there is a single RNAP that is identical in archaea and eukaryotes. Rephrase, or introduce homology between archaeal and eukaryotic RNAPs first

Page 3, line 69: "...to to recruit" word duplication

Page 4, line 82: "TFIIB composed of TBP, the TFIIB-like factors Brf1 and Bdp1 (B double (spelling mistake) prime or B)" The plural "factors" suggests that Bdp1 is TFIIB-like. Change to "factor".

Page 4, line 101ff: "Similarly, the interaction of human TBP with the U6 promoter is characterized by short lifetime in millisecond range while interaction of yeast TBP with a classical RNAP II promoter is highly stable for minutes to hours and bending occurs in two steps."

The life time of yeast TBP with a U6 promoter has also been reported to have a half-life time of 15 minutes (Cloutier et al, PNAS 2001). The sentence should thus be changed to "*(...) yeast TBP with a classical RNAP II or a U6 RNAP III promoter..." in order to clarify that the differences are likely those between yeast and human rather than Pol II and Pol III.

Page 4, line 105: Citation is missing

Page 7, line 183 "withing". Correct spelling

Page 7, line 191: "FRET" abbreviation has been introduced already

Page 10, line 255ff: "In both cases, however, the complex was still force sensitive and only a small fraction (Brf2 31%, TFIIB 30%) of molecules was found in the bent state at 6.6 pN (Figure 2, Supplementary Figure 5".

These results are reported in Figure 3 (not Figure 2) and Suppl. Fig 5 should be better introduced (see above).

Page 11, line 265ff: "In contrast, TFIIB suffices to ensure such a stable complex formation in the Pol II system". The authors report high-FRET populations of 49% for TFIIB-TBP and 69% for Brf2-Bdp1-TBP under high-force conditions. Therefore, the conclusion that TFIIB stabilizes the bent state in the same way as Brf2-Bdp1 is not supported by the data, and the differences should be discussed more accurately.

Page 11, line 258f: "Addition of Bdp1 to the TBP-Brf2-DNA complex, however, did not substantially affect the complex lifetime when linear promoter DNA was used for smFRET measurements⁵²" In the cited study, the life time is reported to increase from 6 to 9 minutes when the Bdp1 SANT domain was added. Is this not considered a substantial increase? In any case, it should be mentioned that the cited experiment was not performed with full length Bdp1 but only with the SANT domain.

Page 14, line 330f "These close-range interactions vary in strength between -0.3 to -8 kcal/mol which falls into the range covered by our experiments". The relationship between binding energy and force should be explained. That kcal/mol and pN fall into the same range is not immediately obvious.

Page 15, line 363 "might not escape from the promoter"

Using the term "promoter escape" for the dissociation of Pol III from TFIIB is misleading and should be rephrased. Promoter escape marks the transition between early, abortive transcription (producing RNAs that only encompass a few nucleotides), and transcription elongation, and it occurs in all three RNAPs.

Page 18, line 405. The term "lineup" is unclear.

Page 17, line 407ff: "Interestingly, extending the initial TBP-DNA complex by additional transcription factors increases the lifetime of the unbent state increases with force". The meaning of this phrase can be guessed, but is hard to understand/incorrect. Rephrase.

Figure 2 and Figure 3: It would help to include the percentage of the high-FRET and low-FRET populations in the figures. Also it is unclear what the grey Gaussian fits in the middle and lower panels represent. Are the reported numbers for the percentage of high-FRET populations the sum of both fitted peaks? This should be explained more clearly.

Reviewer #3 (Remarks to the Author):

The manuscript "DNA origami-based single-molecule force spectroscopy unravels the molecular basis of RNA Polymerase III pre-initiation complex stability" by Kramm and co-workers reports on a clever experimental design revealing the force dependency of the TATA-binding protein binding characteristics and the role of an additional initiation factor Bdp1. The quality of the written manuscript and especially the underlying experimental data is very high. Furthermore, the addressed biological questions are of high relevance improving our understanding of the complex nature of transcription initiation. I therefore recommend the manuscript for publication in Nature Communications especially after the following, mainly minor, comments and suggestions have been addressed.

Main point: The authors use both confocal and camera based single-molecule FRET detection schemes, but it is not directly clear from the figures which modality was used. Please add a note into the figure legend. Following up, comparing the bound FRET peak in figure 3A (TBP + U6, confocal microscopy?) with figure 4D (TBP + U6, TIRF?) reveals a discrepancy in the FRET values of the bound state that needs further explanation ($E = 0.39$ versus $E = 0.75$). From the confocal data one would not expect to see any peak at $E = 0.75$, which has been used for the Hidden Markov Modelling in Fig.4D. In fact, the expected FRET change from the confocal measurements is rather small and would therefore be difficult to be picked up with HMM leading to the question whether the observed reduction of the "bent" life time for the U6 promoter upon increase of the applied force is solid. Could the authors comment?

Figure 1: Could the authors add the number of ssDNA bases used in case of the various reservoir adjustments to the figure? This would make it easier to understand the idea behind the entropical forces.

Figure 2: The authors do not yet state in the legend (but in the supplementary table) how the medium FRET population has been fitted which probably represents temporal averaging. It looks here as it was fit with a third Gaussian (although more extensive frameworks exist that could fit the shape of the entire histogram using a dynamic model see works by Nir/Kapanidis/Seidel). I would also suggest to add the relative occupancies of the bound state (ideally with std determined from bootstrapping or calculated from the three independent measurements) directly to the figure.

Figure 4: Is B representing a single molecule or an histogram of all molecules?

Additional minor points:

- line 69, delete one "to"
- line 82 spelling mistake "double"
- End of introduction: Tune down "we demonstrate for the first time"... soon followed by "that have not been accessible before"
- line 365: reorder references
- line 542 S value is not explained

Response to reviewer comments:

We thank all reviewers for their helpful comments, which helped to improve our manuscript. Our response to the comments is highlighted in blue. Changes in the manuscript are highlighted in red.

Reviewer #1

Kramm et al. investigate in this study the stability of the human Pol III preinitiation complex using single-molecule FRET measurements. They challenge the complex using a previously established origami force-clamp technique and compare the obtained results with the Pol II preinitiation complex. The major finding is that the Pol III specific factor Bdp1 increases the formation of the preinitiation complex at elevated forces from 31% to 69%. At low forces Bdp1 does not appear to impact the complex formation. From this the authors speculate that the Bdp1-induced force resistance is important for RNAP III to cope with torsional as well as topological stress from the surrounding chromatin, which is, however, not further tested. I have a bit mixed feelings on this manuscript. Overall the experiments are carefully executed. I am, however, not really competent in judging how important the found Bdp1-induced force resistance on the Pol III preinitiation complex is for the transcription field. Regarding the biophysics of the preinitiation complex formation, I find it a pity that force-dependencies were measured but not further related to known conformational changes of the promoter DNA.

Detailed re remarks:

The second half of the introduction is quite hard to follow since it covers the preinitiation complexes of RNAP II and III for archaea, yeast and humans. For non-transcription experts this provides a quite hard time when reading. I suggest to focus the introduction more towards the topic of the study.

In the revised version of the manuscript, we deleted information about the archaeal system to clean up the introduction.

As shown in many previous biophysical investigations, a force-dependence of the equilibrium of a binding reaction can be quantitatively related to an underlying conformational change along the force direction (e.g. early work of the Wuite lab on restriction enzymes). This is barely discussed in the manuscript and also not quantitatively explored. Since the force is applied on the DNA ends, a force dependence would be expected if the distance between the DNA ends is shrinking during the binding step. In my opinion this can explain the force-dependence of all association reactions where the FRET shifts to higher values. In contrast for the additional Bdp1 association no change of the DNA end distance is seen from the FRET measurements, which can explain the absent force dependence.

We thank the reviewer for this thoughtful comment and now include a reference to the earlier work of the Wuite lab (Van den Broek et al., NAR 2005, 33, 8, 2676-2684). We agree with the reviewer that the equilibrium should be force dependent if the underlying conformational change is along the force direction. Assuming a Boltzmann distribution, we used the force dependent population change to calculate the expected DNA end-to-end distance change (assuming a constant force during the conformational change. This is justified due to the properties of the force clamp as described in the SI of ref. 63). For the TBP/AdMLP system we calculated a distance change of 1.2 nm. In comparison, we calculated a distance change of the FRET pair of 2.8 nm. The deviation is explained by the different location of interrogation. As the reviewer points out correctly, the force is applied at the DNA ends of the double stranded region whereas the FRET pair reports on distance changes in the centre of the DNA.

Towards the ends the DNA will react to the applied force and comply by about 0.4 nm (as obtained from a Young Modulus of DNA of 1 GPa). In addition, it is known that the binding of the transcription factors causes torsion of the DNA that induces uncertainty in the FRET quantification along the DNA (together with the well-known limitations of trying to deduce quantitative distances from FRET data). Another source of error could arise from slight changes in the exact distance of the suspension points of the ssDNA in the origami force clamp. We are currently studying this effect to derive an even more precise description of the DNA origami force clamp (we expect corrections of a few percent) which is subject of a future publication.

Interestingly, we find a weaker force dependence for TFIIB binding to the DNA/TBP system which also exhibits a smaller conformational change along the force direction indicating that the argumentation of the reviewer might be applicable to this system. A quantitative analysis of these multiprotein systems is not possible as we cannot resolve the different fractions of the different subpopulations including DNA, DNA/TBP and DNA/TBP/TFIIB. What clearly remains is that some proteins have a strong influence on the force resistance of the complex (such as Bdp1) whereas others do not (e.g. TFIIA).

We have included a short paragraph in the manuscript indicating this line of argumentation as well as its limitation for quantification. We also added a Supplementary Figure 6 in the supporting information with the fit yielding the distance change from the force dependent changes of the binding.

In my opinion this would even suggest that Bdp1 would bind after Brf2?

The reviewer is correct with this assumption. It has been shown that Bdp1 is the last factor that enters the TFIIB complex and confers resistance to heparin and high ionic strength.

The presented kinetic analysis of the preinitiation complex formation is somewhat puzzling to me. From the force dependence of the rates the authors suggest that TBP uses different mechanisms when binding to the AdML compared to the U6 promoter. I have a hard time to understand this given that it is the same protein in both cases such that a similar mechanism would be expected.

We agree that these data are surprising. However, we observed this behaviour repeatedly with different protein preparations, different single-molecule setups, different experimentators and even in different labs so that this difference in kinetics is a consistent finding over the years. As the only difference is the sequence of the promoter in these experiment (only 4 out of 8 bases match between the promoters), we sought to understand the difference between these promoters investigating i) the intrinsic curvature of the DNAs (using freely available software tools, e.g. <https://www.lfd.uci.edu/~gohlke/dnacurve/>); ii) based on structural data, we investigated whether TBP forms less direct contacts to the bases of the DNA in case of the U6 promoter, iii) whether the promoter differ in their base stacking energies. We did not find any significant difference for all the criteria listed above. Hence, there is a factor that contributes to this biological system that we unfortunately cannot rationalise yet. Nevertheless, the stabilising contribution of additional transcription factors can be nicely quantified and again agree with data from our own and other labs.

Can the authors relate the different force dependence of binding and dissociation to DNA conformational changes?

According to the X-ray structures (see Supplementary Figure 11 and PDB: 5IYB and 5N9G), TBP invokes the same degree of bending in both promoters. Hence, the extent of conformational change is not a varying factor for the TBP-DNA complexes and should not influence the kinetics.

Alternatively, can they exclude that the different applied techniques were causing the difference in force dependence?

Based on our observations, it seems unlikely that the techniques used for the calculation of the complex lifetimes influence the force-dependency. E.g., i) we initially tried to determine the lifetime of the complex assembled on the AdMLP using TIRF-based single-molecule measurements (as we have done for the U6 promoter). We observed stable TBP-DNA complex that remained in the high FRET state for extended periods of time (e.g. minutes) and rarely observed a transition to the low FRET state. The observation time in TIRF-measurements is limited by the photostability of the fluorescent dyes and consequently, we could not record fluorescence signal for minutes to hours. However, the observation that the TBP-DNA complex is very stable and rarely dissociates over minutes once it is formed is congruent with the lifetime determined using confocal measurements (as shown in Figure 4A/B) and previous measurements with TBP from yeast (doi: 10.1093/nar/gku273). ii) the kinetic data confirm the overall observation that the bent state is stabilised in the presence of additional transcription factors. This observation is independent of the technique used to determine the lifetime of the bent state in the presence of TBP or TFIIA and/or TFIIB and Brf2 or Brf2/Bdp1. Hence, we conclude that the kinetic measurements are truthfully reporting on the biological system.

After checking the Supplement I found that the kinetic analysis was also performed for the more complete forms of the preinitiation complexes. This was not mentioned in the main text. Can one draw any helpful conclusion from this additional data?

We followed the suggestion of the reviewer and mention these data in the revised version of the manuscript. We also added the overview plot of all kinetics measurements (**Supplementary Figure 9F**) to **Figure 4** as panel **E**).

There are a few typos in the manuscript, e.g.:

line 62: "angle of 90°C" (:-))

line 69: 2x "to"

We thank the reviewer for pointing out these mistakes. We corrected them.

line 100: "TFB" should be "TFIIB"?

This part of the manuscript was deleted to increase the readability of the text (see comment above).

Reviewer #2

The conclusion that TFIIB suffices to form stable Pol II promoter complexes is probably not generally correct, but rather depends on the promoter. In fact, in their control experiment in which the U6 promoter was used for measurements with the Pol II factor TFIIB, TFIIB and Brf2 show identical levels of stabilization. However, this experiment is not properly introduced in the text. This experiment also shows that the authors measure the difference in the “bendability” between the U6 and the AdMLP promoter rather than differences between TFIIB and Brf2. The authors should better introduce the experiments and results shown in Suppl. Fig. 5 and discuss them more extensively.

The reviewer is correct in pointing out that TBP in combination with TFIIB can bind the U6 promoter and confer a stabilising effect similar to the stabilising effect observed with Brf2. Please note that we had to use 10-fold higher concentrations of TFIIB to invoke the same stabilising effect as Brf2: TFIIB was used at 200 nM and Brf2 at 20 nM (TBP 20 nM in both cases). This argues for a more efficient stabilising mechanism of Brf2. Moreover, one has to compare the stabilising effects of TFIIB and Brf2 when using the promoters they are binding in the cell: at higher forces, the stabilising effect of TFIIB is much more pronounced in case of the AdMLP (e.g. measurements at 6 pN; AdMLP: 50% high FRET vs U6: 30% high FRET). Bending of the DNA is mainly invoked by TBP and structural studies show that both complexes adopt the same bending angle suggesting that the difference in sequence does not result in different “bendabilities”. The differences in stabilisation efficiency of TFIIB and Brf2 on the U6 promoter can be rationalised by the missing BRE element in the U6 promoter that is required for efficient association of TFIIB with the TBP-DNA complex. Following the reviewer’s suggestion, we included a description of these results in the results section and highlight the differences in concentrations in the results section and the figure legend of **Supplementary Figure 5** and **9**. However, we did not discuss these data as they do not contribute to the overall main message of the manuscript especially considering that the new TFIIA data were added to the manuscript that require discussion.

The biological novelty of the manuscript is limited. It has been previously demonstrated that Bdp1 confers exceptional stability to the Brf2-Bdp1-TBP-DNA complex. Furthermore, because of the issues listed in point 1) it is difficult drawing general conclusions about the role of TFIIB and Brf2 in forming stable promoter complexes. The novelty therefore lays more in the application of DNA origami-based single-molecule force spectroscopy to compare the force sensitivity of Pol II and Pol III promoter complexes.

Recent structural studies have established that the Bdp1 subunit adopts a similar position in the Pol III pre-initiation complex as TFIIA in the Pol II system. In agreement with this, TFIIA has been demonstrated to stabilize the TBP-TFIIB complex on suboptimal promoters. To further increase the impact of the study, it would be interesting to see if addition of TFIIA to the TBP-TFIIB system can confer similar stabilization as Bdp1 does in the Pol III system

We followed the suggestion of the reviewer and included TFIIA in our studies. The new data can be found in **Figure 2** and **Supplementary Figure 9**. In summary, the data show that TFIIA stabilises the TBP-DNA complex. However, in contrast to Bdp1, the stabilisation is not force-resistant and not additive (e.g. at high forces, presence of TFIIA and TFIIB does not confer an increased stability as compared to TFIIB only).

The interpretation of the results in the discussion section is based on a model in which Pol III stays permanently bound to the promoter during transcription. It should be kept in mind that this remains a hypothetical model. Noteworthy, the cited structure of Pol III bound to TFIIIB and an elongation scaffold (Han et al, Cell Discovery, 2018) lacks RNA, indicating that this might be an in vitro artefact through the use of a mismatched DNA bubble. It is also unlikely that the entire transcribed region of a Pol III gene can be accommodated in the active site as stated at page 15, line 373ff. Lastly, up to three Pol III enzymes transcribing the same gene have directly been visualized by electron microscopy in yeast (French et al, MCB 2008), showing that not all transcribing Pol III enzymes are tethered to the promoter. In conclusion, the authors should tone down their statement that Pol III remains always bound to the promoter, but rather state that Pol III might be tethered to the promoter through an interaction with TFIIIB during elongation in some cases.

We adjusted our wording to avoid an over-interpretation of the data in the revised version of the manuscript.

Page 3, line 34: "intiation". Correct spelling

Page 3, line 55: Change "phase" to "phases"

Page 3, line 69: "...to to recruit" word duplication

Page 4, line 82: "TFIIIB composed of TBP, the TFIIIB-like factors Brf1 and Bdp1 (B double (spelling mistake) prime or B)" The plural "factors" suggests that Bdp1 is TFIIIB-like. Change to "factor".

Page 4, line 101ff: "Similarly, the interaction of human TBP with the U6 promoter is characterized by short lifetime in millisecond range while interaction of yeast TBP with a classical RNAP II promoter is highly stable for minutes to hours and bending occurs in two steps." The life time of yeast TBP with a U6 promoter has also been reported to have a half-life time of 15 minutes (Cloutier et al, PNAS 2001). The sentence should thus be changed to "(...) yeast TBP with a classical RNAP II or a U6 RNAP III promoter..." in order to clarify that the differences are likely those between yeast and human rather than Pol II and Pol III.

Page 7, line 183 "withing". Correct spelling

Page 7, line 191: "FRET" abbreviation has been introduced already

We thank the reviewer for pointing out these errors. We corrected all listed mistakes. We moreover changed the wording in line 101ff as suggested by the reviewer.

Page 2, lines 32-33: "The TATA-binding protein (TBP) and a transcription factor (TF) IIB-like factor compound the fundamental core of all eukaryotic initiation complexes." This does not seem to be the case for Pol I were TBP is dispensable in vitro, as also later stated by the authors.

The role of TBP for transcription initiation in the Pol I is still under discussion. While TBP is dispensable *in vitro*, data have shown that TBP is associated with the Pol I promoter *in vivo* and stimulates transcription from the rRNA promoter in the presence of the upstream activating factor UAF (see doi:10.1074/jbc.273.50.33795 and personal communication with the Tschochner lab). TBP might act in a

different fashion in the Pol I system, though. Following the suggestion of the reviewer, we rephrased this passage and it now reads: “The TATA-binding protein (TBP) and a transcription factor (TF) IIB-like factor are important constituents for all eukaryotic initiation complexes.”

Page 4, line 105: Citation is missing

We did not find a statement that is missing a citation.

Page 3, line 56: The term “archaeal-eukaryotic RNAP” suggests that there is a single RNAP that is identical in archaea and eukaryotes. Rephrase, or introduce homology between archaeal and eukaryotic RNAPs first

Following the suggestion of reviewer#1, we omitted the archaeal part in the introduction to avoid confusion and to increase readability.

Page 10, line 255ff: “In both cases, however, the complex was still force sensitive and only a small fraction (Brf2 31%, TFIIB 30%) of molecules was found in the bent state at 6.6 pN (Figure 2, Supplementary Figure 5”. These results are reported in Figure 3 (not Figure 2) and Suppl. Fig 5 should be better introduced (see above).

We thank the reviewer for pointing out to us that we should also refer to **Figure 3**, which we do now in the revised version of the manuscript. We furthermore elaborate on more detail on the data shown in **Supplementary Figure 5** (please see reply to the previous comment of reviewer 2).

Page 11, line 265ff: “In contrast, TFIIB suffices to ensure such a stable complex formation in the Pol II system”. The authors report high-FRET populations of 49% for TFIIB-TBP and 69% for Brf2-Bdp1-TBP under high-force conditions. Therefore, the conclusion that TFIIB stabilizes the bent state in the same way as Brf2-Bdp1 is not supported by the data, and the differences should be discussed more accurately.

The reviewer is right and we changed this passage to now read: “In contrast, TFIIB (even in the presence of TFIIA) cannot reach comparable levels of stabilisation (only 53% of the molecules are found in the bent state) rendering the minimal initiation complex of the RNAP III transcription system more stable than its RNAP II counterpart.”

Page 11, line 258f: “Addition of Bdp1 to the TBP-Brf2-DNA complex, however, did not substantially affect the complex lifetime when linear promoter DNA was used for smFRET measurements⁵²” In the cited study, the life time is reported to increase from 6 to 9 minutes when the Bdp1 SANT domain was added. Is this not considered a substantial increase? In any case, it should be mentioned that the cited experiment was not performed with full length Bdp1 but only with the SANT domain.

We clarified, which Bdp1 variant was used. The sentence now reads “Addition of the Bdp1 SANT domain (residues 241-396, in contrast to the version with residues 130-484 used in this study) to the

TBP/Brf2/DNA complex did increase the complex lifetime by 43% when linear promoter DNA was used for smFRET measurements.” to clarify that a different Bdp1 construct is referenced here. Moreover, we deleted “... did not substantially affect the complex lifetime ...” and provide the accurate relative change in complex lifetime

Page 14, line 330f “These close-range interactions vary in strength between -0.3 to -8 kcal/mol which falls into the range covered by our experiments”. The relationship between binding energy and force should be explained. That kcal/mol and pN fall into the same range is not immediately obvious.

We edited the passage to include the conversion “... falls into the range covered by our experiments ($1 \text{ kcal/mol} = 4.18 \text{ pN}\cdot\text{nm}$) (Supplementary Figure 10).” We hope this edit in combination with Supplementary Figure 10, which shows the free energy of all analysed complexes in kcal/mol, clarifies the relation between our data and the reference.

Page 15, line 363 “might not escape from the promoter”. Using the term “promoter escape” for the dissociation of Pol III from TFIIIB is misleading and should be rephrased. Promoter escape marks the transition between early, abortive transcription (producing RNAs that only encompass a few nucleotides), and transcription elongation, and it occurs in all three RNAPs .

We rephrased this passage and it now reads “... might not disengage from the promoter during transcription elongation ...”.

Page 18, line 405. The term “lineup” is unclear.

We rephrased this passage and it now reads: “our studies show that the addition of Bdp1 to the RNAP III initiation complex is necessary to maintain an active initiation complex ...”

Page 17, line 407ff: “Interestingly, extending the initial TBP-DNA complex by additional transcription factors increases the lifetime of the unbent state increases with force”. The meaning of this phrase can be guessed, but is hard to understand/incorrect. Rephrase.

We thank the reviewer for pointing this out to us. This sentence was deleted from the manuscript.

Figure 2 and Figure 3: It would help to include the percentage of the high-FRET and low-FRET populations in the figures. Also it is unclear what the grey Gaussian fits in the middle and lower panels represent. Are the reported numbers for the percentage of high-FRET populations the sum of both fitted peaks? This should be explained more clearly.

We edited the figures and added the relative occupancy of unbent and bent state in figure 2 and 3. The medium FRET population is now explained in more detail in the material methods section (see response to reviewer #3 below)

Reviewer #3

The authors use both confocal and camera based single-molecule FRET detection schemes, but it is not directly clear from the figures which modality was used. Please add a note into the figure legend.

We added “... on diffusing molecules ...” to indicate confocal solution measurements and “... from TIRF experiments ...” to the figure legends of Figure 2, 3 and 4.

Following up, comparing the bound FRET peak in figure 3A (TBP + U6, confocal microscopy?) with figure 4D (TBP + U6, TIRF?) reveals a discrepancy in the FRET values of the bound state that needs further explanation ($E = 0.39$ versus $E = 0.75$). From the confocal data one would not expect to see any peak at $E = 0.75$, which has been used for the Hidden Markov Modelling in Fig.4D. In fact, the expected FRET change from the confocal measurements is rather small and would therefore be difficult to be picked up with HMM leading to the question whether the observed reduction of the “bent” life time for the U6 promoter upon increase of the applied force is solid. Could the authors comment?

The data shown in Figure 3 are based on confocal experiments and the data shown in Figure 4D is based on TIRF experiments. The confocal data are corrected for crosstalk, direct excitation, gamma and beta, whereas the TIRF data shown is uncorrected. To evaluate the discrepancy between the bent state FRET efficiencies pointed out by the reviewer, we calculated the correction factors for crosstalk, direct excitation and gamma for the TIRF data yielding a mean unbent state FRET efficiencies of $E=0.14$ for both forces and a bent state FRET efficiency of $E=0.55$ for the 0 pN samples and $E=0.51$ for the 6.6 pN samples (**Supplementary Figure 8**). (confocal data $E=0.15$ and $E=0.39$). As recently demonstrated in a comprehensive single-molecule FRET study that compares confocal and TIRF-based FRET data from different labs (doi: 10.1038/s41592-018-0085-0), FRET efficiencies calculated from confocal and TIRF-based measurements using the same sample differ, which might – in some instances - be a result of surface effects.

Figure 1: Could the authors add the number of ssDNA bases used in case of the various reservoir adjustments to the figure? This would make it easier to understand the idea behind the entropical forces.

We edited Figure 1 panel C to include the length of the ssDNA spring and it now reads: “The length of the ssDNA spring for low (AdMLP: 3.3 pN / U6 promoter: 2.6 pN) and high forces (AdMLP: 6.0 pN / U6 promoter: 6.6 pN) is shown. The slight difference in forces for the 141 nt spring originates from the different dsDNA length of the two promoters (see **Supplementary Figure 1**).”

Figure 2: The authors do not yet state in the legend (but in the supplementary table) how the medium FRET population has been fitted which probably represents temporal averaging. It looks here as it was fit with a third Gaussian (although more extensive frameworks exist that could fit the shape of the entire histogram using a dynamic model see works by Nir/Kapanidis/Seidel). I would also suggest to add the relative occupancies of the bound state (ideally with std determined from bootstrapping or calculated from the three independent measurements) directly to the figure.

We thank the reviewer for pointing out that the fitting and origin of the medium FRET population are not adequately explained in the methods section. We updated this part and the section now reads:

“The data were binned (bin size = 0.025) and the mean value for the triplicate of experiments was plotted as E histogram (see Supplementary Figure 5 for mean histograms with standard deviation). The histograms were fitted with a single (DNA) or triple Gaussian fit due to a medium FRET density ($E = 0.31$ to 0.51) that we observed in all origami experiments involving proteins, but not in control measurements with linear promoter constructs. Therefore, we assume that this effect has no connection to the biological system. To account for this density during data analysis, we added an additional gaussian with a fixed area to all our fits to improve the overall fit quality (see Supplementary Table 4) but did not include it in the results and discussion sections.”

In order to assess whether the population represents dynamic averaging, we performed burst variance analysis (BVA) on all data sets, and none showed dynamic behaviour. This either means that the dynamics are faster than the time scale accessible to BVA or that the population is static. Additionally, the population seems to be specific to the origami measurements as we do not observe a middle FRET population when using linear promoter DNA (panel A in the figure shown below). Based on these observations, we assume that the population has no relevance to the biological system and might be caused by dimerised DNA origamis (see the original publication that introduce the DNA origami force clamp, DOI: 10.1126/science.aah5974). Therefore, we accounted for this population by adding a third gaussian in order to improve the overall quality of the fit (compare two Gaussian Fit in panel B with a three Gaussian Fit in panel C).

Reviewer Figure 1: A medium FRET density is specific to DNA origami-protein complexes. (A) FRET efficiency histograms of HsTBP binding to the AdMLP either fused into a 0 pN DNA origami force clamp (black) or as a linear double stranded promoter (blue). Confocal single-molecule measurements were performed with freely diffusing DNA-protein complexes. (B) The histogram of the 0 pN origami from (A) fitted with a double and (C) triple Gaussian fit.

Furthermore, we performed photon distribution analysis (PDA) on datasets for the AdMLP to test whether this approach yields a different area distribution than our current analysis. Overall, the PDA-derived fits show the same trend as our previous analysis. The area of the low FRET population increases with force as the high FRET population decreases. However, the PDA fits generally yield a smaller medium FRET population (on average 0.06 relative area versus 0.16 in our manual fit). This additional density is primarily re-distributed to the high FRET population (20% increase). As the PDA did not differ

drastically from the analysis performed in the first version of the manuscript, we did not change the analysis method and presentation of the data in the revised version of the manuscript.

Regarding the reviewer's suggestion to show data from three independent experiments in Figure 2: we show averaged data of at least three independent experiments. The mean histogram was fitted to a Gaussian distribution. We added panel C and D to Supplementary Figure 5 to show the variance of the data across the three technical replicates, which demonstrate the consistency of the measurements across the independent experiments.

Additional minor points:

- line 69, delete one "to"
- line 82 spelling mistake "double"
- line 365: reorder references

We thank the reviewer for pointing out these errors and corrected them in the revised version of the manuscript. .

- End of introduction: Tune down "we demonstrate for the first time"... soon followed by" that have not been accessible before"

Following the suggestion of the reviewer, we deleted the phrases to avoid an over-statement.

- line 542 S value is not explained

In the revised version of the manuscript, we refer at this point to a (newly added) section in the Supplementary Information that provides adequate equations for the calculation of the E and S values.

REVIEWERS' COMMENTS:

Reviewer #1 (Remarks to the Author):

Kramm et al. have addressed part of my concerns, e.g. they added an analysis about the force dependence of the system. I overall appreciate the introduction of the DNA origami system to study the force dependence of transcription initiation. I am still not fully convinced how meaningful the concluded differences in the force dependencies are. The concluded force dependence of the TBP-TFIIA-TFIIB system relies finally on a single point (6.0 pN) and it is a bit surprising to see that one gets the same efficiencies as for TBP-TFIIB at 0 pN and at 6 pN, with the only difference being the 3.3 pN value. This questions in my opinion somewhat how reliable one can determine the binding efficiencies (e.g. how significant are the differences at 3.3 pN?). Similarly, the main argument for a force independence of the binding of the TBP-BRF2-Bdp1 complex is (compared to TBP-TFIIA-TFIIB) the final point at 6.6 pN.

Overall, I am still left with some mixed feelings about the manuscript and would actually like to follow the final recommendation of reviewer 2 who seems to be a "transcription expert" who would best be able to evaluate the merits of this study for the field.

Reviewer #2 (Remarks to the Author):

The authors have well addressed the comments of the three reviewers through changes in the text and by including additional experiments. The revisions have considerably strengthened the manuscript. I now recommend publication in Nature Communications.

Reviewer #3 (Remarks to the Author):

In the revision, the authors correct their TRIF data for crosstalk, direct excitation, etc and end up with a corrected E value of $E=0.51$ and 0.55 , respectively. The authors cite a publication, which actually shows that the ΔE is smaller than ± 0.05 between labs and measurement modes meaning that their difference towards 0.39 as measured on confocal is still rather high. Anyways, I leave it here by the mentioning the discrepancy. I don't think it changes any (biological) conclusions drawn from the data.

I further have to take the word of the authors that BVA and PDA analysis did not hint any (fast) dynamic switches (no data shown).

In conclusion, the authors sufficiently addressed my concerns and remarks. I therefore continue to recommend the publication of the manuscript in Nature Communications.

We thank the reviewers for their helpful comments and suggestions throughout the reviewing process. As far as we can see, there are no additional comments and requests from the reviewers that need to be addressed.

Reviewer #1 (Remarks to the Author):

Kramm et al. have addressed part of my concerns, e.g. they added an analysis about the force dependence of the system. I overall appreciate the introduction of the DNA origami system to study the force dependence of transcription initiation. I am still not fully convinced how meaningful the concluded differences in the force dependencies are. The concluded force dependence of the TBP-TFIIA-TFIIB system relies finally on a single point (6.0 pN) and it is a bit surprising to see that one gets the same efficiencies as for TBP-TFIIB at 0 pN and at 6 pN, with the only difference being the 3.3 pN value. This questions in my opinion somewhat how reliable one can determine the binding efficiencies (e.g. how significant are the differences at 3.3 pN?). Similarly, the main argument for a force independence of the binding of the TBP-BRF2-Bdp1 complex is (compared to TBP-TFIIA-TFIIB) the final point at 6.6 pN.

Overall, I am still left with some mixed feelings about the manuscript and would actually like to follow the final recommendation of reviewer 2 who seems to be a "transcription expert" who would best be able to evaluate the merits of this study for the field.

Reviewer #2 (Remarks to the Author):

The authors have well addressed the comments of the three reviewers through changes in the text and by including additional experiments. The revisions have considerably strengthened the manuscript. I now recommend publication in Nature Communications.

Reviewer #3 (Remarks to the Author):

In the revision, the authors correct their TRIF data for crosstalk, direct excitation, etc and end up with a corrected E value of $E=0.51$ and 0.55 , respectively. The authors cite a publication, which actually shows that the ΔE is smaller than ± 0.05 between labs and measurement modes meaning that their difference towards 0.39 as measured on confocal is still rather high. Anyways, I leave it here by the mentioning the discrepancy. I don't think it changes any (biological) conclusions drawn from the data.

I further have to take the word of the authors that BVA and PDA analysis did not hint any (fast) dynamic switches (no data shown).

In conclusion, the authors sufficiently addressed my concerns and remarks. I therefore continue to recommend the publication of the manuscript in Nature Communications.